# Public perception of scientists: Experimental evidence on the role of sociodemographic, partisan, and professional characteristics

Burak Sonmez[1]*, Kirils Makarovs[2], Nick Allum[3]

1 Social Research Institute, University College London, London, United Kingdom, 2 Faculty of Social and Behavioural Sciences, University of Amsterdam, Amsterdam, The Netherlands, 3 Department of Sociology, University of Essex, Colchester, United Kingdom

☯ These authors contributed equally to this work.
* b.sonmez@ucl.ac.uk

**Data Availability Statement:** All data, code, and materials used in the analysis are made available

## Abstract

Previous research shows that public trust in scientists is often bound up with the messages that they convey and the context in which they communicate. However, in the current study, we examine how the public perceives scientists based on the characteristics of scientists themselves, irrespective of their scientific message and its context. Using a quota sample of U.S. adults, we investigate how scientists' sociodemographic; partisan; and professional characteristics affect preferences and trust towards them as a scientific adviser to local government. We find that scientists' party identification and professional characteristics appear to be prominent to understand public preferences towards them.

## Introduction

Understanding how citizens perceive and engage with science is becoming a matter of increasing concern for public policy. The extant body of research in this field is voluminous, but some key themes clearly emerge. Broadly, much of this research has examined civic scientific literacy [1], attitudes to science [2], public engagement in science policy- shaping [3, 4], as well as exploring public sentiment around polarising issues e.g. global warming [5] or vaccinations [6]. However, the central role in this field is assigned to the concept of trust in science [7, 8]. Science as a public enterprise cannot prosper without public trust. Despite hot-button issues such as global warming and, more recently, COVID-19, activating partisan conflicts, the best evidence we have shows that overall, the U.S. public expresses a high and stable level of confidence in the scientific community compared to other institutions [9]. That said, the American public is both culturally and politically divided in their support for science [10]. For instance, even though politically conservative people have positive attitudes towards scientific research, they hold negative attitudes towards the scientific community [11].

Previous research is limited to draw causal inferences to understand how certain characteristics of scientists that are likely to make them appear more or less preferable or trustworthy amongst the public, despite the extant literature on the perception of scientists and its

via Open Science Framework (OSF) repository. DOI.ORG/10.17605/OSF.IO/Y572W.

**Funding:** N.A received funding from the University of Essex as part of his personal research account. https://www.essex.ac.uk/. The funders had no role in study design, data collection and analysis, decision to publish, or preparation of the manuscript.

**Competing interests:** The authors have declared that no competing interests exist.

predictors [12, 13]. For instance, what we do know is that the majority of Americans tend to regard scientists as intelligent, honest, and focused on solving real-world problems [14]; that scientists working for universities rather than government or industry are perceived as more trustworthy [15]; and that there is a gradient of authority related to the specialisation within which scientists work [16].

Looking beyond these broad perceptions of scientists is necessary in order to understand the public opinion dynamics of particular controversies. Scientists play a pivotal role in communicating the contents of their research to a wider audience and informing policymakers about the implications of their discoveries [17]. Trust in scientists has been found to shape public opinion about science controversies [18] and mediate how exposure to mass media translates into beliefs about global warming [19], and it can positively affect public trust in science media [20].

Despite enjoying a high level of public trust in broad terms, the scientific community can face a struggle with signaling the integrity of research to a general audience. One of the reasons for this is that the institutionalised instruments that underpin research integrity–e.g. study pre-registration, funding disclosure, peer-reviewing–that are used to indicate adherence to accepted codes of scientific conduct are not always transparent and uniformly interpreted by general audiences [21]. The other–even more fundamental reason–is the perceived social distance between scientists and the public [22]. The divide is reinforced by keeping laypeople outside of the dialogue on science policy [23] and cultivating the image of scientific research as monolithic and not prone to any kind of 'human' indeterminacy and frailty [24]. As a consequence, Fiske and Dupree [25] show that as much as scientists are viewed as competent and intelligent, they are also perceived as 'low-warmth' professionals who are distant from the general public and thus oftentimes unable to ignite trustworthiness and convey the credibility of their research. This status quo hampers the dissemination and legitimisation of scientific outcomes and does not facilitate the convergence of opinion between scientists and the public. However, a question that has been somewhat overlooked is whether scientists themselves should be treated as a homogeneous group or whether public preference for scientists is heterogenous based on their sociodemographic, partisan, and professional characteristics.

We aim at addressing this gap by studying how the characteristics of scientists themselves contribute to how they are perceived by the public, irrespective of the scientific message conveyed–a message that may itself trigger heterogeneous reactions that colour perceptions of the scientists themselves.

## Understanding public preferences for scientists

Social identity theory [26–28] posits that individuals navigate through the social world by categorising people into groups, namely into those to whom they feel closeness and similarity (in-group) and those who are perceived as distant and even threatening to them (out-group). One way to maintain a positive social identity for an individual is to demean the out-group by stereotyping some qualities of its members in an intentionally exaggerated way. Stereotypes, hence, serve as a cognitive shortcut that allows people to engage with motivated reasoning–making quick judgements without spending cognitive resources on conscious scrutiny [29]. In this paper, we examine 5 distinct characteristics of scientists that we believe are the most pertinent when it comes to looking at what might shape people's preferences for scientists. These characteristics are in detail sex; race/ethnicity; place of work; field of studies; and political identification.

## Sociodemographic characteristics

Many stereotypes about gender roles degrade women and deny their possession of traits historically deemed exclusive to men, such as high-level intellectual abilities [30] or strong leadership competencies [31]. Moreover, women are still often expected to prioritise family and care-giving over professional achievements and to engage with household duties more than men [32]. Academia is also subjected to the sex disparities [33]. Male students outnumber female students in the number of science majors at college, even though up to high school both sexes enroll in roughly the same number of math and science courses [34]. As a consequence, women tend to be underrepresented in science, especially in the STEM field. The latest National Science Foundation (NSF) Indicators report suggests that although women constitute 52% of the college-educated workforce in 2018, they account for less than a third of all the scientists working in the fields of engineering (16%), computer science and mathematics (27%), and physical science (29%) [35]. With these considerations in mind, our first expectation is that female scientists will be perceived as less preferable and trustworthy than male scientists (Hypothesis 1).

In addition, there are negative racial stereotypes attached to African-American and Hispanic populations, who tend to score lower on science literacy than whites, even when educational attainments are accounted for [36, 37], and they feel more alienated from science overall [38]. The negative racial stereotyping has been shown to induce stress among African-Americans when they are undertaking tasks framed as tests of cognitive abilities [39]. In that vein, people coming from minority ethnic backgrounds are often subjected to prejudice regarding their intellectual abilities and capacity for work [40, 41]. Given the racialised stereotypes, it is not surprising that ethnic minorities are underrepresented in scientific employment [42]. For instance, the latest NSF data shows that ethnic minorities make up only between 10% to 22% of the total science and engineering workforce in the U.S. in 2018 –most dramatically underrepresented in life sciences (10%), physical sciences (11%), and engineering (12%) [35]. Hence, we think that racialised perceptions of scientists are another salient factor to investigate.

This brings us to the hypothesis that Black and Hispanic scientists will be perceived as less preferable and trustworthy compared to white (Hypothesis 2.1). The opposite set of stereotypes surrounds the identity of Asian Americans. In line with the notion of 'model minority' [43] attributed to Asians, suggesting a generally positive appraisal of this group, Asians are perceived as more industrious; academically successful; [44] and nerdy [45] than whites and other minorities. They are viewed as more likely to succeed in such intellectually demanding jobs as engineering; computer science; and mathematics [46]. Hence, we expect that Asian scientists will be perceived as more preferable and trustworthy compared to whites (Hypothesis 2.2).

## Professional characteristics

Levels of public trust in institutions vary depending on whether they are government, industry or academic. Scientists working in these domains may be viewed differently too. The NSF data highlights that 40% of the U.S. population has a 'great deal of confidence' in the scientific community in 2016, while half this fraction trusts in major companies (18%); organised labour (13%); the executive branch of the federal government (13%); and congress (6%) [47]. There is a long debate about the impact of market efficiency and private profits on the integrity of the scientific enterprise [48]. The opponents of the commercialisation of research claim that it can lead to widely occurring conflicts of interests; restricted public access to the benefits of research; and constraints on the circulation of ideas within the broader scientific community [49]. Privately funded scientists may be motivated more

by external (profit) rather than intrinsic (public good, benevolence) factors, thus compromising the credibility of their scientific claims [50].

On the other hand, scientists working for governmental research institutes, albeit being less vulnerable to accusations of profit-related self-interest, potentially face mistrust from a different perspective. Public confidence in the government has been gradually eroding from the 1960s up until now [51]. One of many potential reasons for this is a growing perception that important information is concealed from the public, leading to the emergence of alternative narratives e.g. about 9/11 [52] or the John F. Kennedy assassination [53] or more recently QAnon [54]. Nevertheless, universities, despite facing increasing demands for transparency; accountability; and public participation [55], are still regarded as the most credible producers of scientific knowledge as far as the public is concerned. Therefore, we propose the hypotheses that scientists affiliated with industry will be viewed as less preferable and trustworthy (Hypothesis 3.1) and those working for universities will be viewed as more preferable and trustworthy (Hypothesis 3.2) than scientists working for the government.

There is a rich scholarship on how scientific disciplines engage in boundary-work [56, 57] to delineate themselves from each other and from non-science, thus legitimising knowledge claims, attracting more resources and gaining influence in the public sphere [58, 59]. Gauchat and Andrews examined how the public maps sciences according to their 'cultural authority', that is to say, the perceived scientific prestige of the field in relation to the institutions and actors holding political and economic power [60]. They show that while hard sciences such as physics, medicine, and biology tend to score high on the dimension of scientific prestige, the U.S. public does not perceive economics as a very scientific discipline [16].

Some fields are prone to permeation by conspiratorial theories. Contentious topics such as vaccination [61], infectious diseases [62], climate change [63]–and their respective scientific fields (e.g. environmental and medical sciences)–are more likely to be contaminated by conspiratorial narratives and undermined as sources of valid and indisputable knowledge than, say, physics whose authority is rarely questioned. Also, medical and environmental sciences are often characterised by how closely adjacent they are to relevant policy decisions that get embroiled in politics. In particular, policies related to medical and environmental regulations can be perceived as influenced by political biases and agendas, thereby undermining public confidence and credibility of their respective scientific fields.

Given this, we hypothesise that physicists and those affiliated with environmental and medical sciences will be viewed as more preferable and trustworthy than economists (Hypothesis 4.1), and physicists will be viewed as more preferable and trustworthy (Hypothesis 4.2) than those affiliated with environmental and medical sciences.

## Partisanship

Party identification (partyID) could be used as a perceptual screen to quickly identify trustworthy actors [25]. Relatedly, partyID is known to affect how people process information and make inferences from data [64]. The idea of motivated [65, 66] or identity-protective [67] cognition boils down to the notion that when facing controversial questions that are in principle to be processed via objective and unbiased scrutiny, people are rather inclined to shape their answers and attitudes in a way that shields their core beliefs and identities from being threatened. Thus, if scientists visibly endorse a particular political party, people tend to apprehend the content of those scientists' messages differently depending on their own political orientation. The ongoing debate on climate change serves as a clear example of motivated thinking [68]. Empirical research shows that increased levels of education [5] and scientific literacy [69] lead to more acceptance of climate change among Democrats, yet do not change opinions

among Republicans. Similar narratives can be found in the discussion on vaccination behaviour [70]. Also, when scientists address politically polarised scientific issues, people are more likely to perceive the scientists as being a member of the political group associated with that specific issue [71].

Hence, building upon the idea of affective polarisation that partisans favour co-partisans but disfavour opposing partisans [72], we hypothesise that people will more likely prefer those scientists who express the same political stance as they do, e.g. Democrats will more likely prefer and trust those scientists who identify as Democrats, and Republicans will more likely prefer and trust Republican scientists (Hypothesis 5.1). Additionally, the effect of partisanship for scientists working for the government or universities may differ from that which operates for those working for commercial organisations, since Republicans and Democrats have contrasting views on institutions. For instance, a majority of Republicans believe that colleges and universities have a negative impact on the country, while Democrats support the opposite view [73]. However, we are not able to identify established empirical evidence on this potentially unique within-design interaction effect in the literature, thereby keeping the following hypothesis as exploratory rather than confirmatory. Therefore, we would like to explore whether the effect of scientist's partisanship interacts with their place of work (Hypothesis 5.2).

## Materials and methods

We adopt a conjoint survey experiment design, which is extensively used in marketing research to reveal multidimensional consumer preferences [74]. This survey experimental design had little traction in social and political science until Hainmueller, Hopkins, and Yamamoto established a formal model linking the conjoint design to the Neyman-Rubin causal model, without invoking strong model assumptions [75]. Following their canonical study, conjoint experiments have been designed to study various topics, such as immigration preferences [75]; the social construction of illegality [76]; attitudes towards asylum seekers [77]; and more frequently, candidate preferences [78, 79]. Unlike traditional factorial survey experiments, conjoint experiments are designed to optimise the capacity to decompose the effects of multidimensional traits simultaneously. More precisely, a conjoint design allows researchers to disentangle the effects of multiple causal factors on subjects' preferences, choices or ratings over distinct candidates or stimuli through hypothetical scenarios. Hainmueller and colleagues argue that conjoint experiments can measure individuals' choices in real-world situations, comparing their experimental estimates to behavioural benchmarks in real-world outcomes to ensure external validity [75].

Given the methodological advantages outlined above, we regard the conjoint design to be well suited for studying public perceptions of scientists, who are often presented by media to the public on the basis of their personal characteristics and professional affiliations. The main advantage of this design is that it allows respondents to compare two scientists on each characteristic simultaneously and encourage them to more thoroughly engage with the information about the scientists. Thus, in our conjoint design, we created a scenario where respondents are asked to compare and judge the profiles of hypothetical scientists for the role of scientific advisor to local government, in five rounds, who vary along multiple dimensions, such as sex, race/ethnicity, scientific field, place of work, and in some randomly selected cases, political party identification. Table 1 displays a full list of characteristics in the dimensions. S1 Appendix also provides an example of the scenario (paired profiles). Overall, this design permits us to estimate the relative contribution of a set of scientists' characteristics to their perceived favourability without being confounded by other contextual factors.

**Table 1. Characteristics of conjoint profiles.**

| Characteristics | Scientist A (Levels) | Scientist B (Levels) |
|---|---|---|
| **Race/Ethnicity** | {White, Black, Hispanic, Asian} | {White, Black, Hispanic, Asian} |
| **Sex** | {Male, Female} | {Male, Female} |
| **Political Party Affiliation** | {Democrat, Republican, Independent} | {Democrat, Republican, Independent} |
| **Scientific Field** | {Physics, Medical science, Environmental science, Economics} | {Physics, Medical science, Environmental science, Economics} |
| **Domain of Work** | {University, Government, Industry} | {University, Government, Industry} |

The experiment was designed in Qualtrics and performed between March 18[th], 2020 and March 20[th], 2020 after the study was preregistered at the Open Science Framework and granted ethical approval at the University of Essex (ETH1920-0447), with 1005 U.S. adults recruited through Prolific, which provided us with a high quality representative (quota) sample based on age, sex, and ethnicity [80]. S1 Appendix also provides the demographic characteristics. All questions and question blocks were randomly ordered to avoid spill-over effects. Although the time period of data collection corresponded with the early stages of the Covid-19 pandemic, we do not have reason to believe that in March 2020, attitudes towards scientists had changed much from their pre-pandemic positions. According to surveys from Pew Research, declines in trust in scientists, and in medical scientists in particular, only began after April 2020 [81]. That said, it is important to acknowledge that there was a partisan gap in the confidence in the U.S. health care system in dealing with the coronavirus. That is, in early March 2020, a vast majority (87%) of Republicans had confidence in the U.S. health care system to handle the response to the coronavirus, while a slight majority (53%) of Democrats had this positive view [82]. This partisan heterogeneity in turn may have a differential impact on respondents' confidence in medical science and potentially scientists in this scientific field. Hence, it is crucial to include measures of respondent's partisanship in our study design, which can capture the main currents of difference between Republican and Democrat respondents.

In designing our experiment, we followed the methodological suggestion that relatively small sample sizes (less than 1000); low number of trials; and high number of levels (5 or more) may lead to severely under-powered designs [83]. Taking these considerations in account, our study satisfactorily reaches the conventional power threshold (above 0.80)–the calculations were made through the R Shiny application by Stefanelli and Lukac [83].

We combine a choice-based and rating-based conjoint designs [84] (see S1 Appendix, S1 Fig 1), in which two scientist profiles are presented next to each other. The choice-based framework requires participants to sequentially compare sets of scientists' profiles and for each set pick one scientist that they prefer the most. Our first main outcome variable specifically measures participant's scientist preference, which is asked "Which of these two scientists do you prefer to elect as the member of the Board of Scientific Councillors in your district?". While choice-based decision is relatively easy to process for the participants as it mimics a context of real-world elections [79], it does not elucidate participants' indifferent position in decision-making.

Using the rating-based outcome variables can shed light on the decision-making process by asking participants to 'give a numerical rating to each profile which represents their degree of preference for the profile' [84]. In this study, we additionally use three rating-based outcome variables that enable us to show how perception of scientists is also linked to trustworthiness of scientists and particular trust. Several characteristics that are commonly associated with trustworthiness in a trustee have been identified, including ability, benevolence, and integrity in the integrative model of organisational trust [85]. More precisely, the integrative model conceptualises ability encompassing a wide array of skills, competencies, and personal

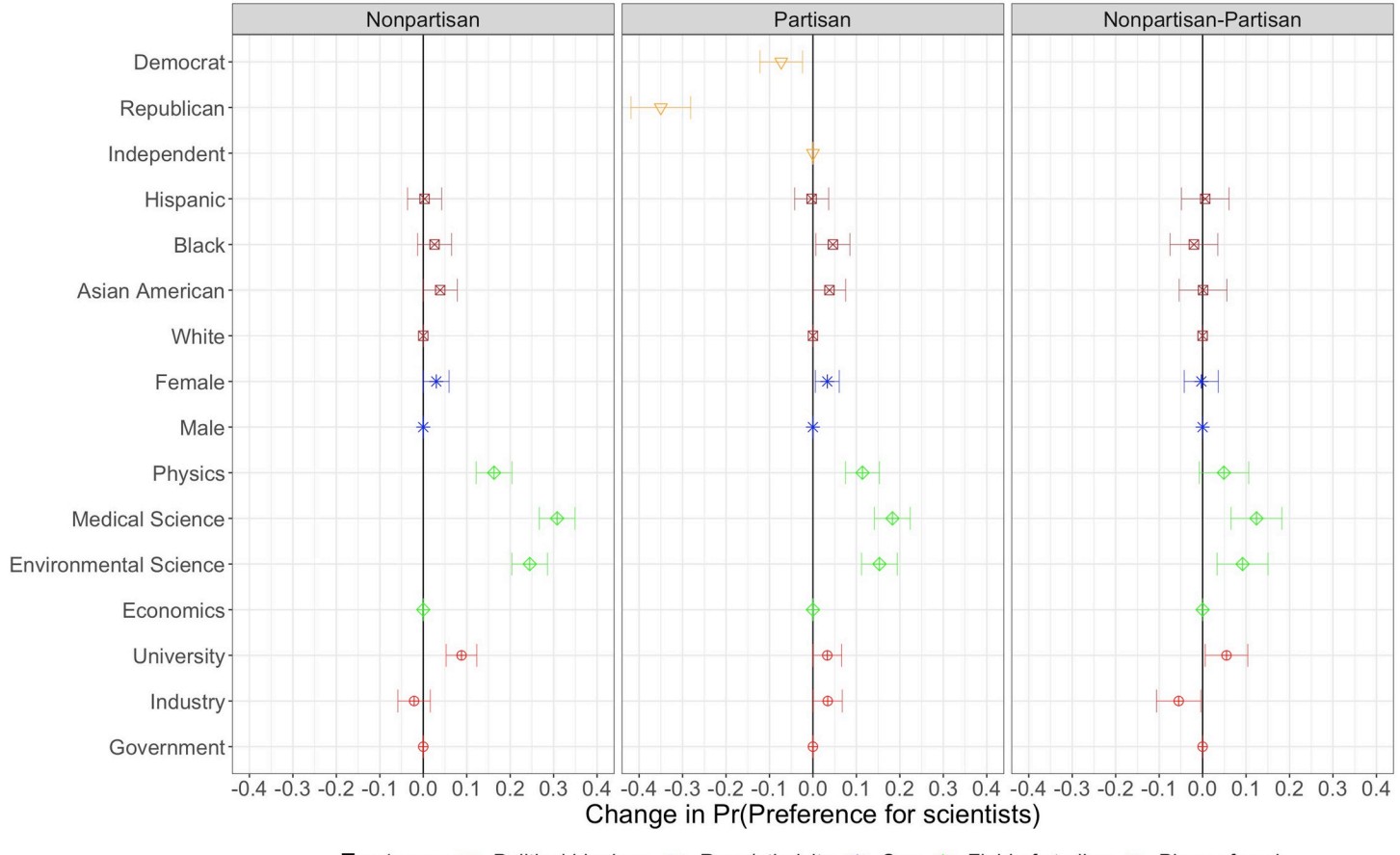

**Fig 1. Estimated average marginal component effects (AMCEs).** Dots represent point estimates of AMCEs that show the average effect of a particular attribute value against another value of the same attribute in a scientist profile on the probability of choosing a particular scientist. The first panel (Nonpartisan) shows the results of the conjoint profiles without the partisan profiles. The second panel (Partisan) shows the results of the conjoint profiles with the partisan profiles. The third panel (Nonpartisan-Partisan) shows the differences between the partisan and nonpartisan profiles. Reference categories are: Independent, White, Male, Economics, and Government. Segments represent their 95% confidence intervals. Standard errors are clustered at respondent level. $N_{individuals} = 1005$ and $N_{observations} = 10050$.

characteristics that enable an individual or collective to exert a measurable degree of influence within a specific domain. The specificity of the domain in question is a crucial consideration, as it serves to highlight the specialised nature of the ability in question. For example, an individual who possesses a high degree of technical acumen may be deemed to possess a high degree of trustworthiness in tasks and responsibilities related to that technical domain. This is due to the notion that the possession of such specialised knowledge and skills lends credibility to the individual's ability to competently execute said tasks. Relatedly, benevolence and integrity are considered as normative dimensions of trustworthiness in this model, as they both play a crucial role in determining the level of trustworthiness that can be placed in a trustee. Benevolence, for instance, refers to the extent to which the trustee is believed to act in the best interest of the trustor, without any self-serving motivations. Integrity, on the other hand, is to evaluate the consistency between a trustee's actions and their sense of justice. These dimensions are often cited in literature and are seen as a comprehensive set that helps explain a significant portion of trustworthiness. In light of such dimensions of trustworthiness, in this study we assume that scientists can earn public's confidence in their qualifications and fairness when conducting research.

Building on the literature using similar dimensions in explaining trustworthiness of experts and scientists, we define trustworthiness of scientists as a multi-dimensional concept composed of epistemic and normative considerations. That is, we specify epistemic trustworthiness as those futures of scientists, that make laypeople depend on their ability or competence and defer to them due to own limited resources and knowledge [86]. Hence, the first rating-based outcome variable that we use is 'Where would you place your assessment for this candidate's qualifications?'. It is measured on a 7-point scale where 7 is highly qualified and 1 is highly unqualified. This item taps into the expertise dimension of epistemic trustworthiness, measuring scientist perceived level of knowledgeability and overall competence. The second rating-based outcome variables represents the fairness [12] dimension of normative trustworthiness by referring to scientist's moral character and goodwill. Participants are specifically asked: 'How much would you agree that this potential candidate cares about the best interests of the public when conducting his/her research?'. This item is measured on a 7-point scale where 7 is strongly agree and 1 is strongly disagree. Lastly, we invoke a concept of particular trust, which can be viewed as an antipode to generalised trust [87]. While generalised trust is abstract and does not presuppose any tangible trustee, particular social trust 'is associated with specific people or groups of people, whether known or in-group others'. The concept of particular trust is especially relevant in the context of conjoint experiments, where participants have to make a round of decisions about specific distinct profiles of people. We measure particular trust in scientists by asking participants 'How much would you trust Scientist A and B?'. This last item is also measured on a 7-point scale where 7 is strongly trust and 1 is strongly mistrust.

Research subjects may use cognitive shortcuts in evaluating profiles, for instance, over-weighting the first characteristic shown to them or some particularly salient attributes. Hence, we randomised the order of the characteristics for each pair of profiles. Last but not least, drawing on the same strategy as adopted by Kirkland and Coppock and Sen [79, 80], in order to directly test the salience of partisanship and its impact on other characteristics, a randomly half of respondents were shown the scientists' political party identification in the paired profiles while the others were not. In analysing the effects of our experimental manipulations on these four dependent variables, we follow an alternative non-parametric estimation strategy as suggested by Hainmuller and colleagues [75]. That is, the causal quantity of interest is called the Average Marginal Component Effect (AMCE) rather than the Average Treatment Effect (ATE), which precisely represents how much the probability of choosing a scientist profile would change, on average, if one of the scientist's characteristics (e.g. scientific field) was changed from one level to another (e.g. from economics to physics). The AMCEs can effectively be estimated without bias, using a following linear regression model:

$$y_{ijk} = \alpha + \sum_{l=1}^{5} \boldsymbol{\beta}_l' x_{ijkl} + \epsilon_{ijk} \tag{1}$$

where $y_{ijk}$ is respondent i's preferences for scientist $j \in \{1,2\}$ in the kth round ($k \in \{1, 2, 3, 4, 5\}$); $\alpha$ is an intercept; $x_{ijkl}$ is a vector of dummy variables showing that scientist j in the kth round of respondent i trusted or mistrusted; $\beta_l$ is a vector of those dummy variables' coefficients; and $\in_{ijk}$ represents individual-level idiosyncrasies, an error term. We make the required assumption that the errors are independent of each other and of scientist characteristics, which is justified by the experimental design. Since the respondents are given two profiles to evaluate in five rounds, standard errors are clustered by respondent to avoid biased estimates of the variance. We separately estimate Eq 1 based on two subsets of the sample, in which scientists' dimension of party identification is randomly added to the list of characteristics for some respondents. In addition, we condition the estimation on the research subject's political

identification to detect heterogeneous treatment effects. In doing so, Leeper, Hobolt, and Tilley [88] shed light on an important point that conditional AMCEs can be substantially misleading when interpreting the degree of favouring or disfavouring between subgroups, since interactions are sensitive to the baseline category used in regression analysis [84]. In other words, they demonstrate that the size and the direction of differences-in-AMCEs have negligible relationship to the underlying degree of favourability of the subgroups toward profiles with certain features. Following this, the baseline category choices can make similar preferences look dissimilar and dissimilar preferences look similar. Given that, we should interpret AMCEs as the difference in the size of the casual effect for groups, but not as a way of descriptively characterising differences in preferences between the groups. Therefore, as suggested by Leeper, Hobolt, and Tilley [88], we also provide marginal means of both our main results and subgroup preferences by respondent's party identification, since reporting marginal means enables us to demonstrate differences for all levels of characteristics without being interpreted relative to the baseline categories [84]. Overall, this provides us with an additional presentation of differences between group preferences.

## Results

We present average marginal component effects (AMCEs), as well as report marginal means. First, we separately present the effects of scientists' characteristics for those respondents who were randomly selected to receive and not to receive the attribute of scientists' political party identification in the conjoint profiles. Second, we examine how the attribute of place of work is moderated by scientist's political party identification. Third, we report conditional AMCEs and marginal means to examine potential heterogeneous treatment effects of scientists' characteristics by respondents' political party identification. Last but not least, we test the effects of scientists' characteristics on the extent to which they are deserved to be trusted along three dimensions of trust.

In column one of Fig 1 the effects of changes in characteristics of scientists on the probability of preferring for scientists are shown without the party identification attribute, whose presence in conjoint profiles randomly varies across respondents. The results indicate that respondents clearly prefer scientists who are specialised in natural sciences by a margin of 16–31 percentage points (see also S1 Appendix, S1 Table 3), relative to scientists in economics, corroborating Hypothesis 4.1.

In contrast to our expectation for Hypothesis 4.2, scientists affiliated with physics are favoured less compared to those from medical and environmental science (see Fig 2). The results also support Hypothesis 3.2, that respondents are more likely to prefer scientists working at universities to scientists working for the government, by a margin of 9 percentage points (see also S1 Appendix, S1 Table 3). Even though respondents slightly prefer Asian American, Black and female scientists relative to white male scientists, these effects are very small (see Fig 1). While the finding confirms Hypothesis 2.2, it does not confirm that male scientists are perceived more favourably, compared to female scientists (Hypothesis 1). When we move to subgroup analysis, we find that Republican respondents are more likely to prefer male scientists, while Democrats favour female scientists (see Figs 3 and 4).

The second column of Fig 1 shows that, overall, respondents who were exposed to the partisanship characteristic, are less likely to prefer partisan scientists relative to politically independent scientists. We examine heterogeneity in these treatment effects by partisanship later. Additionally, once respondents know the political orientation of scientists, the effects of scientific field and place of work are attenuated. The last column of Fig 1 indicates that respondents statistically significantly alter the marginal weight given to the characteristics of scientific field

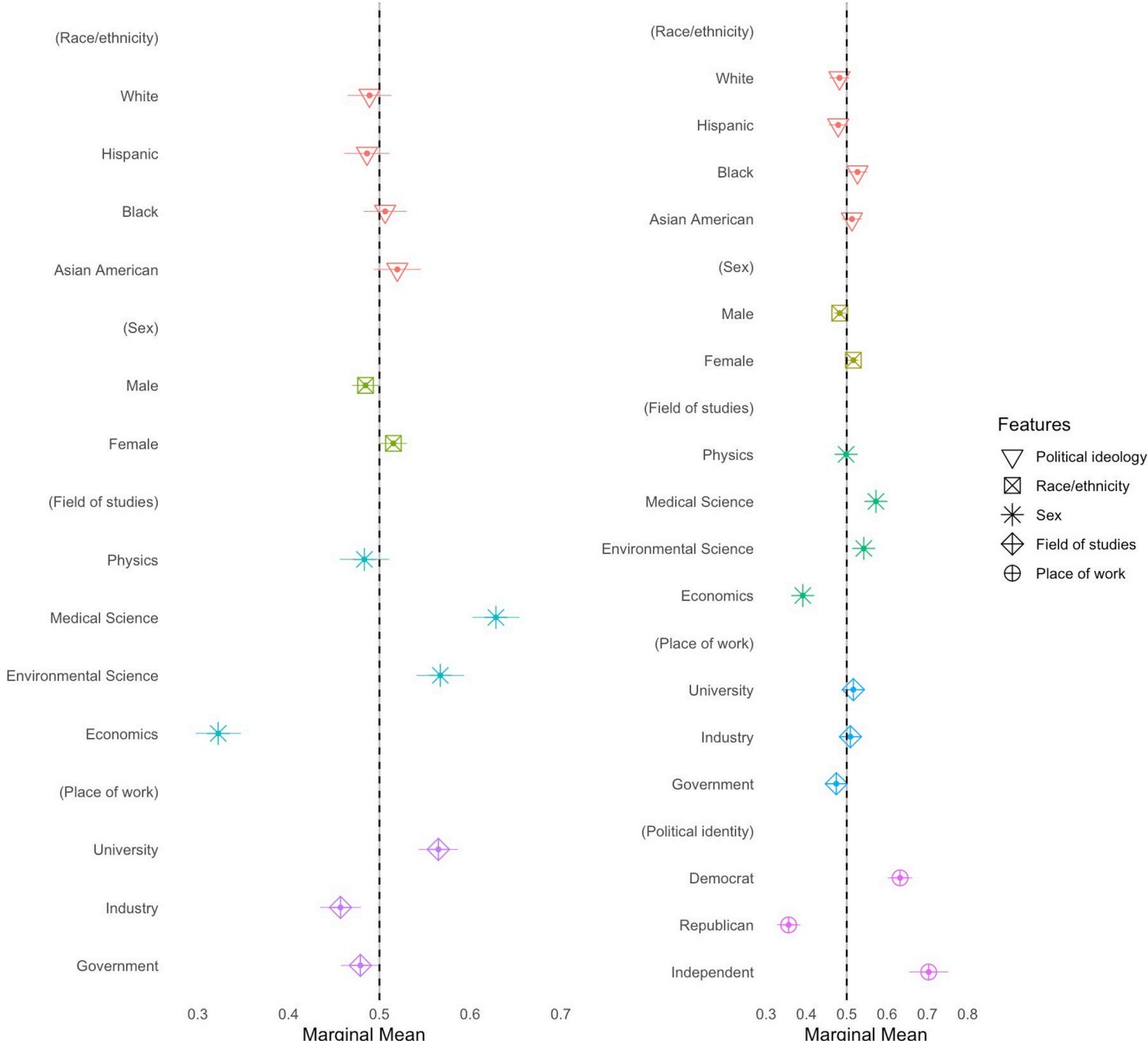

**Fig 2. Marginal means (MMs).** The figure on the left is for the conjoint profiles without the partisan profiles. Marginal means represent the mean outcome across all appearances of a particular conjoint feature level, averaging across all other features. In our forced-choice conjoint design with two profiles per choice task, marginal means have a direct interpretation as probabilities: these MMs average 0.5 with values above 0.5 indicating features that increase scientist's favourability and values below 0.5 indicating features that decrease scientist's favourability. Segments represent their 95% confidence intervals. Standard errors are clustered at respondent level. $N_{individuals} = 1005$ and $N_{observations} = 10050$.

and place of work if they are shown the characteristic of political party identification. By contrast, the difference in AMCEs for other attributes is insubstantial.

Analysing within-design interactions, given the results in S1 Table 10 (S1 Appendix) and Fig 5, in Hypothesis 5.2, we test the exploratory hypothesis that the effect of the characteristic

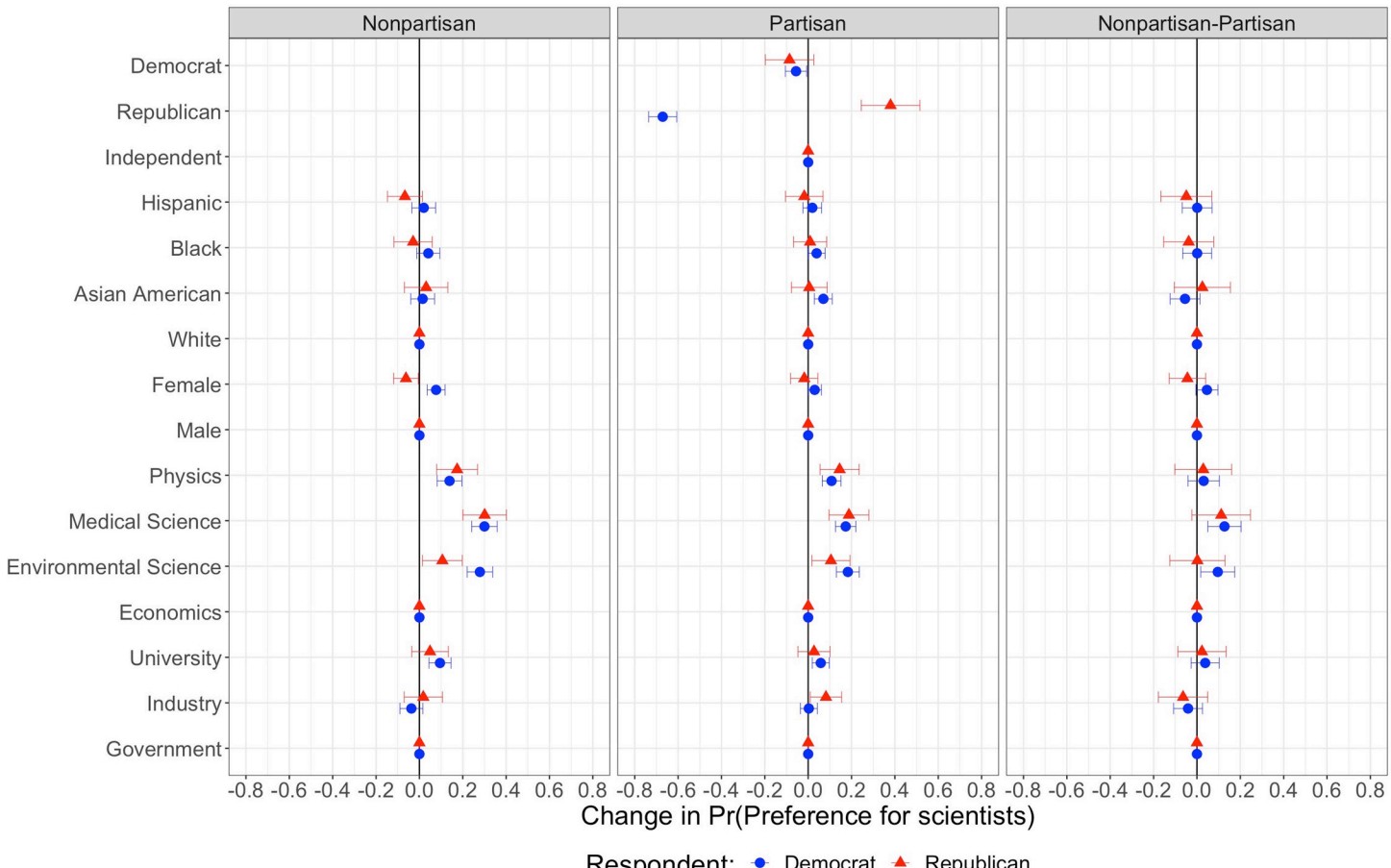

**Fig 3. Heterogeneous treatment effects by respondent's political identification.** Conditional Average Marginal Component Effects (AMCEs). Dots represent point estimates of conditional AMCEs that show the average effect of a particular attribute value against another value of the same attribute in a scientist profile on the probability of choosing a particular scientist, conditional on respondent's political identification. The first panel (Nonpartisan) shows the results of the conjoint profiles without the partisan profiles. The second panel (Partisan) shows the results of the conjoint profiles with the partisan profiles. The third panel (Nonpartisan-Partisan) shows the differences between the partisan and nonpartisan profiles. Reference categories are: Independent, White, Male, Economics, and Government. Segments represent their 95% confidence intervals. Standard errors are clustered at respondent level. $N_{individuals} = 711$.

"domain of work" on favouring a scientist varies across scientist's party identification characteristic. The first three panels represent the estimated AMCEs of scientists' domain of work, conditional on the scientist's party identification. The right most two panels show the average component interaction effects (ACIEs) with respect to party identification and domain of work. Overall, there appear to be no substantively meaningful differences in preferences for the characteristic "domain of work" conditional on scientist's party identification, in reference to the baseline level (being politically independent).

## Heterogeneous effects by respondent's party identification

We reproduce our results conditioning on respondent's own partisanship (see Figs 3 and 4). If respondents are not exposed to information regarding scientists' partisanship, being Democrat increases the preference for female scientists by 8 percentage points (see also S1 Appendix, S1 Table 4), while Republicans are 6 percentage points less likely to choose female scientists. We also observe that both Republicans and Democrats have a pronounced preference for scientists in natural sciences relative to economists, but Democrats substantially favour environmental

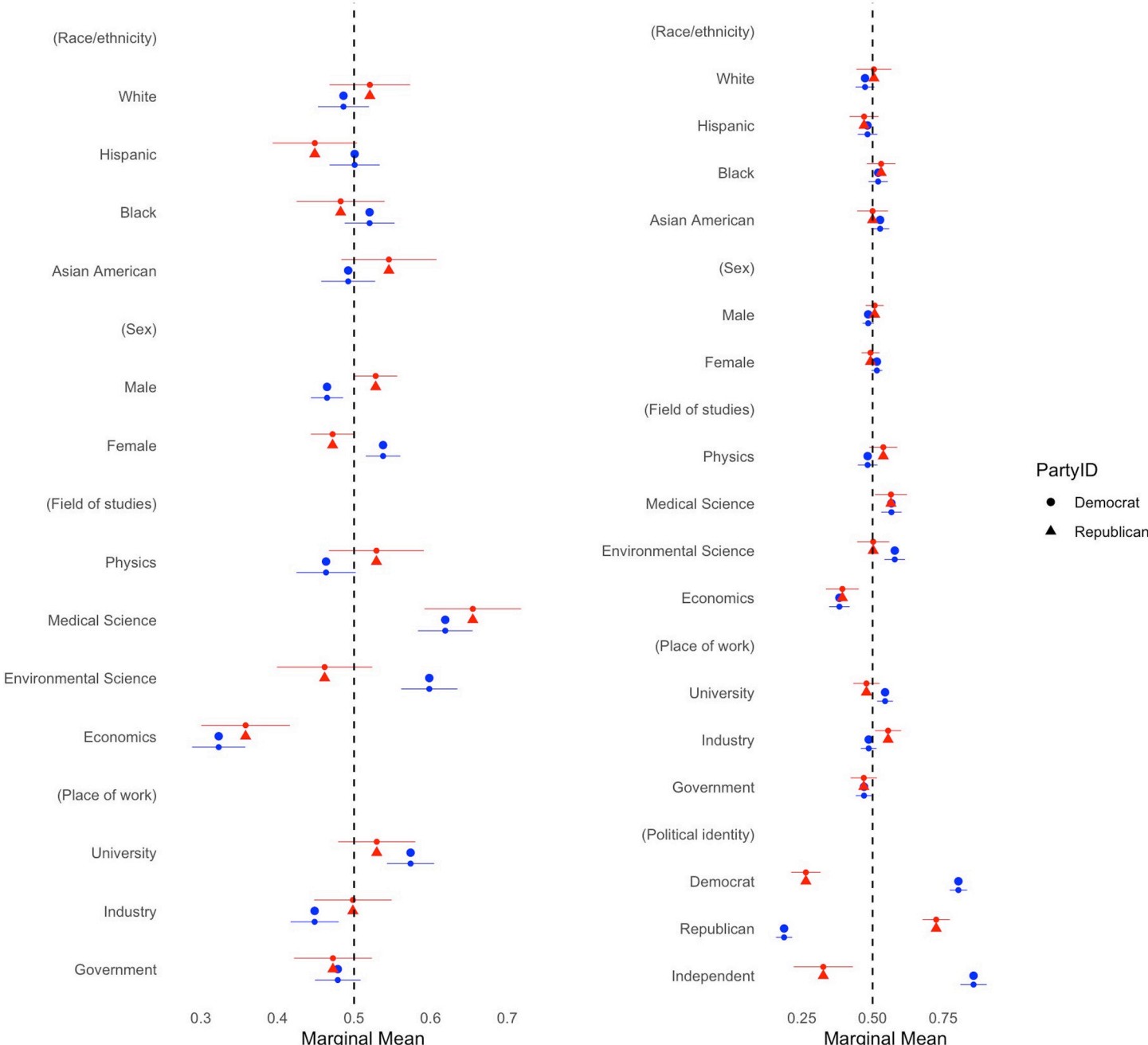

**Fig 4. Marginal means by respondent's political identification.** The figure on the left is for the conjoint profiles without the partisan profiles. Marginal means (MMs) represent the mean outcome across all appearances of a particular conjoint feature level, averaging across all other features. In our forced-choice conjoint design with two profiles per choice task, marginal means have a direct interpretation as probabilities: these MMs average 0.5 with values above 0.5 indicating features that increase scientist's favourability and values below 0.5 indicating features that decrease scientist's favourability. Segments represent their 95% confidence intervals. Standard errors are clustered at respondent level. $N_{individuals} = 711$.

scientists more than Republicans do. Although our main theoretical prediction is confirmed–that respondents prefer scientists working at the universities relative to scientists working for the government–disaggregated results show that this is true only for Democrat respondents, not Republican respondents. In addition, while Democrat respondents equally favour

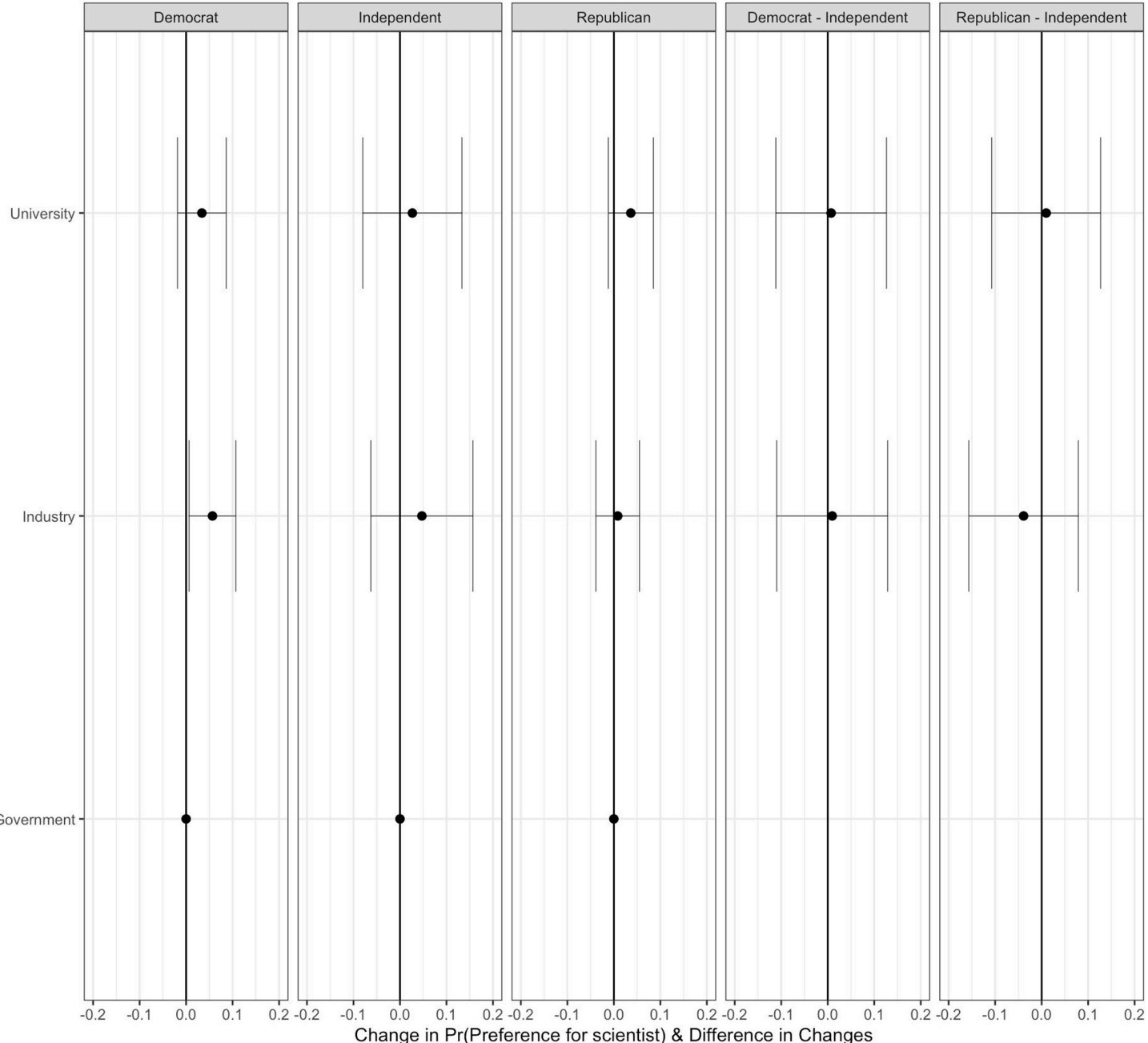

**Fig 5. Average effects of scientists' domain of work on respondents' preference, given the scientist's party identification.** The first three panels represent the estimated AMCEs of scientists' domain of work, conditional on the scientist's party identification. The rightmost two panels shows the average component interaction effects (ACIEs that show the causal effect of scientist's domain of work moderated by scientist's party identification) with respect to party identification and domain of work. Segments represent their 95% confidence intervals. Standard errors are clustered at respondent level. $N_{individuals} = 1005$.

Democrat and politically independent scientists, Republicans only favour scientists who share their political orientation. This finding confirms our Hypothesis 5.1.

The effects of scientists' ethnicity/race and place of work also vary between Democrats and Republicans. Specifically, while scientists in industry are preferred by Republican respondents, Democrat respondents choose scientists at universities over others. Democrats are also 7 and 4

percentage points more likely to favour Asian American and Black scientists respectively, relative to their white counterparts (see also S1 Appendix, S1 Table 5). Both Democrats and Republicans remain indifferent to scientists' ethnicity or race except for the case of Asian American and Black scientists.

## Perception of scientists through dimensions of trustworthiness and particular trust

Fig 6 demonstrates that the majority of our main findings hold across all dimensions of trustworthiness albeit with reduced effect sizes (see also S1 Appendix, S1 Tables 7–9). That is, while respondents find natural scientists more trustworthy by 3–11 percentage points relative to economists, the effects for sex and race/ethnicity are either muted or are negligible across all measures of trustworthiness and particular trust. Given that respondents are less likely to consider partisan scientists trustworthy relative to politically independent scientists across all measures of trustworthiness and particular trust, the effects of partisanship become relatively smaller, but the effects of field of studies become more salient when respondents evaluate trustworthiness of scientists based on their qualifications, namely epistemic trustworthiness. Thus, once scientist's competence is the focal point in determining trustworthiness, the impact of partisanship becomes less salient. Considering the perceived prestige of the scientific fields [16], we confirmed one of our hypotheses that scientists working in the fields of physics, medical and environmental science are perceived as more trustworthy relative to those from economics, especially in the context of epistemic trustworthiness. In line with the main results, we confirm another set of hypotheses that respondents are more likely to consider scientists working at universities trustworthy, relative to scientists working for the government, but they are less likely to consider scientists in the industry trustworthy in context of the normative trustworthiness. Overall, this shows that people view natural scientists more trustworthy than economists, and the gap becomes wider when trustworthiness of scientists is conceptualised based on competences. Also, scientists working for the industry become less trustworthy when trustworthiness is concerned with public interests.

## Discussion

In this research, we have examined how the presence of various characteristics of scientists can shape public perceptions of them through preferability and trustworthiness for a scientific public advisory role. The strength of our research design, using a conjoint analysis, means that we can evaluate the effect of these characteristics simultaneously.

Our results show, rather reassuringly, that professional characteristics, such as the scientific field and type of institution in which scientists work, matter more than their sociodemographic characteristics, such as sex, race and ethnicity. Those working at universities and in natural and medical sciences are favoured over economists and those working in industry or government. This broadly aligns with previous survey findings, where medical and university scientists are amongst the most trusted [16]. Exposure to scientists' partisan sympathies does not alter the importance of demographic features of scientists, but is important in weighing the impacts of scientific field and workplace. Overall, there is a preference for scientists in a local scientific advisory role to be politically neutral. However, we find that when we disaggregate these preferences for Democrat and Republican respondents, we find that both groups show a penchant for scientists sharing their own party ID, although in the case of Democrats, politically independent scientists are marginally preferred. It is noteworthy that political orientation acts as a cue in assessing trustworthiness of scientists despite the fact that in many, if not most, instances, this would have little influence on scientific work. That said, this experiment

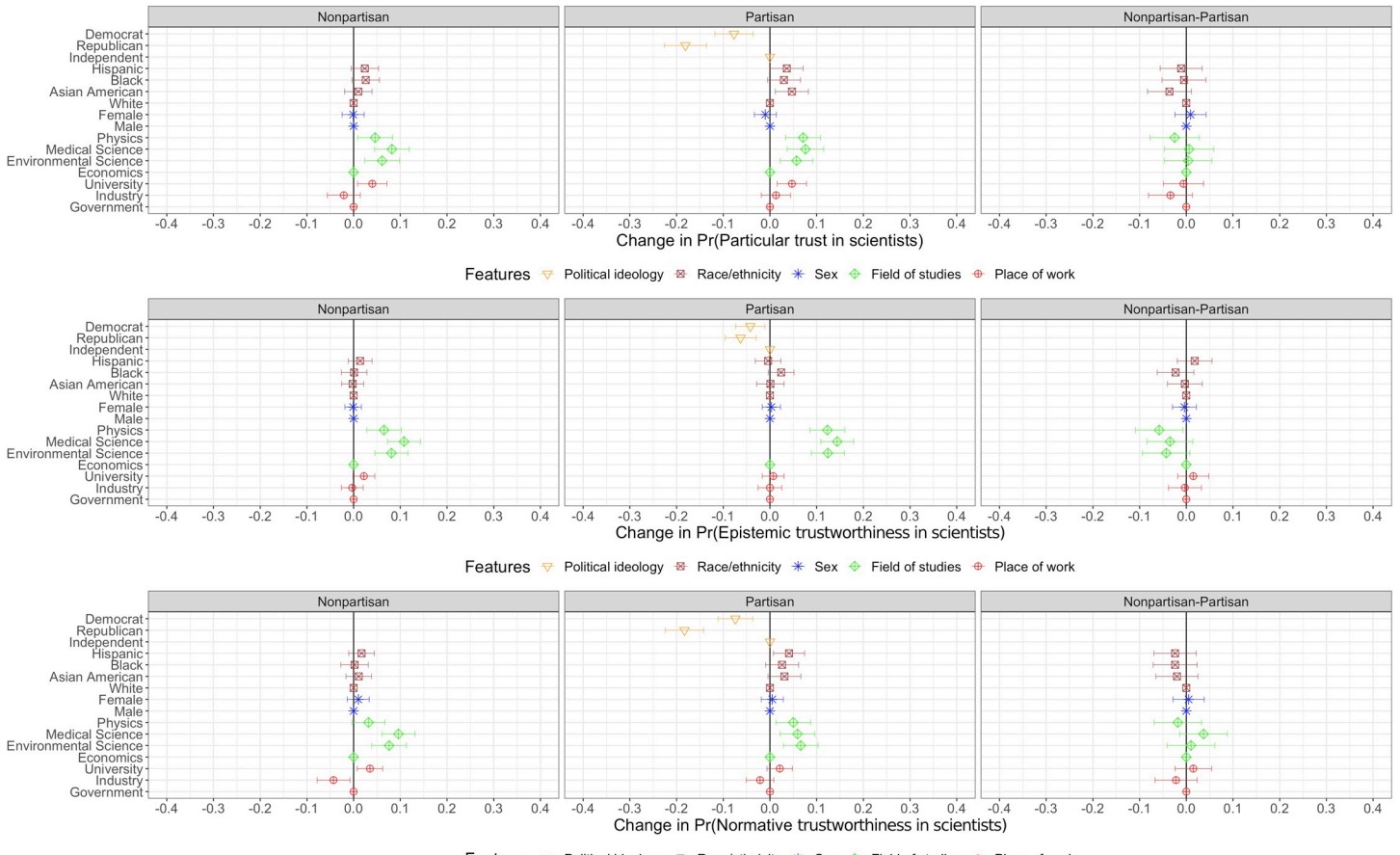

**Fig 6. Multidimensionality of trustworthiness and particular trust.** Estimated Average Marginal Component Effects (AMCEs). Dots represent point estimates of AMCEs that show the average effect of a particular attribute value against another value of the same attribute in a scientist profile on the probability of choosing a particular scientist. The first panel (Nonpartisan) shows the results of the conjoint profiles without the partisan profiles. The second panel (Partisan) shows the results of the conjoint profiles with the partisan profiles. The third panel (Nonpartisan-Partisan) shows the differences between the partisan and nonpartisan profiles. Reference categories are: Independent, White, Male, Economics, and Government. Segments represent their 95% confidence intervals. Standard errors are clustered at respondent level. $N_{individuals}$ = 1005 and $N_{observations}$ = 10050.

asks respondents to choose a scientific advisor to local government, which arguably is a role more open to the influence of politics than basic or applied research.

We should acknowledge that although our research sheds light on how different features of scientists change their perceived trustworthiness in a broader context, further research can identify how trustworthiness of scientists would also change across different domains of expertise through context-specific questions (e.g. to what extent do you think that Scientist A/B has expertise in policy-making?).

Even though our experimental design is well-suited for identifying public perceptions of scientists and has an ability to mitigate social desirability bias [89], we are mindful of the limitations of our research. First, hypothetical preferences in conjoint designs may increase the artificiality of the task. For instance, in our study, respondents take a decision task in which they are asked to elect a scientist for the Board of Scientific Councillors in their district. However, the general public is less likely to be part of such elections in real-world context. The artificial setting therefore undermines the ecological validity of the findings. In addition, the specific language used to define the scientific board in the task (i.e. Board of Scientific Councillors in their district) may have been misunderstood by the U.S. participants, since the concept

is not widely known. This limitation may lead to some unmeasured error in operationalising our dependent variables. The second limitation lies in the operationalization of scientists' political orientation. While measuring it by referring to the well-established categories of political party affiliation (Democrat, Republican, Independent) is appropriate for the US political landscape and conducive to the conjoint design, this is not the only way to do so. Instead of explicitly presenting political affiliation, one could tap into political identification through the proxy alignment of cultural worldview (e.g. see Kahan et al. [69]), or by presenting behavioural cues (e.g. voting in favour of anti-abortion bills) that have a strong linkage to the ideological stance. Further research should explore whether explicit or implicit declaration of scientists' political affiliation affects their perceived credibility and trustworthiness. Finally, our online quota sample is representative of the U.S. population based on demographic variables, such as age, sex, and ethnicity but politically skewed to Democrats. Hence, we used entropy balancing as robustness check to weight our sample in terms of quota indicators: age, sex, ethnicity, and then partisan characteristics (see S1 Appendix for details). The weighted results show that there is no major difference in the preferences for scientists when we adjust our sample to both quota and partisan characteristics (see S1 Tables 11, 12 and S1 Fig 6 in S1 Appendix). On the other hand, we cannot completely eliminate concerns about generalisability of our findings due to the non-probability sampling method used in this research.

Overall, we find that our results contribute to the previous research showing that natural and medical scientists will find it easier to garner trust than social scientists. Specifically, this finding may suggest a need for social scientists to take additional measures to facilitate comprehension of their research by a lay audience. This may include a heightened emphasis on the use of clear and accessible language, the provision of contextual information, and the proactivity in addressing prevalent misconceptions and biases regarding the field. Furthermore, it highlights the need for social scientists to actively engage with the public in order to promote a deeper understanding of their research and to dispel any misconceptions that may exist. In a similar vein, journalists, in their practice of reporting, should be cognizant of the public's perception of the credibility of different fields of science, and critically identify and mitigate any potential biases associated with social sciences in their reporting, through the inclusion of a diverse array of perspectives and scientists in their coverage. Sociodemographic characteristics vis-à-vis professional characteristics of scientists do not appear to act as strongly negative cues, which is contrary to what might have been expected. Consistent with the polarised political landscape in the U.S. more generally, perceptions of partisanship on the part of scientists seem likely to generate differences in the extent to which they are preferred and found trustworthy by different groups, rendering consensus on hot scientific topics more elusive. That said, our findings show that the effect of partisanship (affective polarisation) on trustworthiness of scientists becomes muted when the consideration is epistemic trustworthiness instead of normative trustworthiness and particular trust. This particularly suggests that science communication should draw more attention to those features of scientists that help improve epistemic trustworthiness, especially if the scientific issue at hand is polarised among partisan groups.

## Supporting information

**S1 Appendix.**
(ZIP)

## Acknowledgments

The authors thank participants of the 2019 meeting of ESSEXLab Behavioural Mini Conference for providing their comments on the research design.

## Author Contributions

**Conceptualization:** Burak Sonmez, Kirils Makarovs, Nick Allum.

**Data curation:** Burak Sonmez, Kirils Makarovs.

**Funding acquisition:** Nick Allum.

**Investigation:** Burak Sonmez, Kirils Makarovs.

**Methodology:** Burak Sonmez.

**Project administration:** Burak Sonmez, Kirils Makarovs.

**Software:** Burak Sonmez, Kirils Makarovs.

**Visualization:** Burak Sonmez, Kirils Makarovs.

**Writing – original draft:** Burak Sonmez, Kirils Makarovs, Nick Allum.

**Writing – review & editing:** Burak Sonmez, Kirils Makarovs, Nick Allum.

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
