## [Decision Letter · Decision Letter 0]

12 Dec 2022

PONE-D-22-23245Public Perception of Scientists: Partisan and Non-partisan ThinkingPLOS ONE

Dear Dr. Sonmez,

Thank you for submitting your manuscript to PLOS ONE. After careful consideration, we feel that it has merit but does not fully meet PLOS ONE’s publication criteria as it currently stands. Therefore, we invite you to submit a revised version of the manuscript that addresses the points raised during the review process. Please, read carefully comments from Reviewers 1 and 2 and approach all of them.

We look forward to receiving your revised manuscript.

Kind regards,

Celia Andreu-Sánchez

Academic Editor

PLOS ONE

Journal Requirements:

"The authors thank participants of the 2019 meeting of ESSEXLab Behavioural Mini."

"N.A received funding from the University of

Essex as part of his personal research account. https://www.essex.ac.uk/. The funders had no role in study design, data collection and analysis, decision to publish, or preparation of the manuscript."

Reviewers' comments:

Reviewer's Responses to Questions

**Comments to the Author**

1. Is the manuscript technically sound, and do the data support the conclusions?

Reviewer #1: Yes

Reviewer #2: Partly

2. Has the statistical analysis been performed appropriately and rigorously? 

Reviewer #1: I Don't Know

Reviewer #2: I Don't Know

3. Have the authors made all data underlying the findings in their manuscript fully available?

Reviewer #1: Yes

Reviewer #2: Yes

4. Is the manuscript presented in an intelligible fashion and written in standard English?

Reviewer #1: Yes

Reviewer #2: Yes

5. Review Comments to the Author

Reviewer #1: The technical bit of this seems really interesting and I'm glad someone has tried this. I've had a book sitting in my office on conjoint experiments for a related question. Nevertheless, there's a couple of conceptual things here that I think are really important to integrate before this is published.

1. First, I think the authors need to do an edit in which they are more careful in how the use the words such as trust, trustworthiness, and 'dimension.' In this regard, the manuscript often suggests that expertise and 'fairness' are dimensions of trust but that's not quite right.

Per the widely accepted 'integrative model of organizational trust' (as well as Fiske's BIAS model, to some extent), it's quite important to distinguish between trustworthiness perceptions and behavioral trust. In this regard, behavioral trust involves actually making oneself vulnerable to a trustee. The behavior part is important. Trustworthiness, in contrast, is a word often used to refer to perceptions and thus ability (or expertise, if you use Fiske's terminology, or competence if you use Hovland's), benevolence (an aspect of warmth for Fiske; part of trustworthiness for Hovland), and integrity (an aspect of warmth for Fiske; part of trustworthiness for Hovland) can be called dimensions of trustworthiness but not dimensions of trust. This distinction is obnoxious but it matters because we really need to make sure we're distinguishing between perceptions/beliefs and behaviors when we do trust-related research.

This distinction is also important because it suggests that just asking people 'how much do you trust X' -- as you do -- will give you really messy data as we have little idea what people are actually thinking about when they respond. Even better is if we ask people more specific questions such as 'to what degree do you think X has expertise in ...' or 'to what degree would you be willing to take the advice of X on Y". Beyond just usage, this edit should thus influence how you understand your results. In any case, I just think you need to clean this up.

2. My other ask is that you specifically address the validity/utility of the approach you used. I understand using this type of experiment when you're trying to decide on whether a product with X1 characteristic is more/less appealing than product with X2 characteristics. Indeed, I first learned of the approach in the context of cigarette labeling and it seems reasonable to ask people to pick between two similar options.

What I don't get here is why it make sense to ask someone to pick between two scientists. When would someone ever do that unless they were on a hiring committee? And wouldn't doing that make the differences especially salient? Practically, for most times that we ask people to trust scientists, I don't see why we wouldn't be better doing things like presenting systematically randomly different scientists and asking people to rate them on relevant characteristics (e.g., see the work of Shupei Yuan). Practically, I also can't help believing that your results are contaminated by substantial desirability bias.

3. I'm a little sad that you only varied things over which people have little control. From a communication perspective, I can't do much about my race, gender, or employer. I suppose you could choose a spokesperson for an issue in some cases based on these factors but, in general, the scientist you have is the scientist you get. It is true, however, that a scientist can choose to disclose their political ideology (or religiosity in the case of someone like Katherine Hayhoe or James Watson; or values, more generally, as in the case of of some of Kahan's research), but this isn't exactly a common approach. Training programs (and Hayhoe) often encourage scientists to identify values they share with their audience but I've never heard anyone advocate sharing party ID.

4. I think you need to also be a little more careful about how you talk about motivated reasoning. The differentiation between 'motivated' reason and heuristic reasoning seems important here. I'm not sure you're invoking motivated reasoning here inasmuch as you don't have a stimulus that seeks to make a specific identity salient and then ask people to evaluate something using effortful processing. Instead, you're just asking people to make judgements based on heuristic cues, it seems.

5. The figures need extensive notes that include things like range and the meaning of everything. In this regard, I note that figures generally be understandable without substantial reference to the originating text (at least according to APA).

Ultimately, what I'd like to see happen is a more targeted manuscript that focuses on the potential of the method in this space but recognizes the odd, unrealistic context + danger of desirability bias of the current attempt. I would also like to see something that's much more careful with the trust/trustworthiness language used.

Reviewer #2: Thank you for the opportunity to review this manuscript on how the characteristics of scientists themselves shape public perceptions of scientists. Please note that I do not consider myself to be an expert in conjoint designs and thus my observations focus on the introduction/argument, sample and measurement (rather than analysis), and discussion/implications. I hope the authors find these comments helpful as they continue their work in this space.

1. I appreciated the authors’ thorough literature review and argument that informed the selection of the five characteristics of scientists. With regards to the fields of study, I’m curious why the authors did not assess other social/behavioral sciences beyond economics (e.g., psychology, sociology, communication)? These other disciplines would seem germane.

2. For H5.2, can the authors specify the nature of the interaction effect they expected (i.e., “differ” in what way, as far as directionality and strength of association are concerned)?

3. One could imagine a situation in which the scientist does not affiliate themselves directly with a political party or stance, but instead with being supportive of or opposed to a given behavioral strategy, policy, etc. I realize the authors would not, with these data, be in a position to assess what effects this positioning might have on public trust, but perhaps this is worth considering in future research (and thus could be considered in the discussion; see comment #7). My instinct is that this sort of proxy alignment may be more prevalent than direct alignment/declarations of affiliation with a particular political party or stance (i.e., scientists may have more of an implied vs. overt ideological bent).

4. In the abstract and methods, the authors describe the sample as “representative of U.S. adults,” which is misleading, as Prolific does not maintain a panel created via RDD and/or address-based sampling methods (compared to, say, NORC’s AmeriSpeak panel). It is certainly fair to note the quota sampling.

5. This study was fielded during the early days of the COVID-19 pandemic. This was a rather exceptional time, of course, in which to be evaluating trust in scientists—especially in the U.S., where there is considerable evidence that the pandemic was politicized early and often (see, for example, https://press.princeton.edu/books/hardcover/9780691218991/pandemic-politics and https://read.dukeupress.edu/jhppl/article-abstract/45/6/967/165291/The-Emergence-of-COVID-19-in-the-US-A-Public). Can the authors speak to the implications of the timing of data collection for interpreting and contextualizing study results?

6. With regards to DV measurement, I’m not sure that U.S. participants would know what a “Board of Scientific Councillors” is in “their district.” This seems like UK terminology that might not translate. Was this defined for participants? What assurances do we have that participants did in fact understand this reference? Has this measure been used in other U.S.-based contexts?

7. For all the strengths of this study, it is not without limitations, and those should be acknowledged in what is, at present, quite a short discussion section (e.g., see above note about the sample, which has implications for generalizability). Similarly, at present there is a solitary sentence that speaks to the implications of this research for science communication (lines 473-475). Further elaboration here would be helpful: What are the implications for journalistic practice, specifically; for scientists, in their public-facing work; and so forth?

6. PLOS authors have the option to publish the peer review history of their article (what does this mean?). If published, this will include your full peer review and any attached files.

Reviewer #1: No

Reviewer #2: No

---

## [Author Response · Author response to Decision Letter 0]

26 Jan 2023

Thank you for the invitation to revise and resubmit our manuscript entitled “Public Perception of Scientists: Partisan and Non-partisan Thinking” (PONE-D-22-23245). We are grateful to both reviewers and editors for their detailed feedback and suggestions for revisions. We have substantially edited the manuscript in regards to actionable suggestions and criticisms. In particular, the manuscript text has been revised to clearly articulate the theoretical concepts of trust and clarify methodological limitations alongside their implications. We think that the revised manuscript is now much improved in light of this feedback. 

As requested, we detail how we have addressed each of the criticisms and acted upon the suggestions. We have highlighted specific changes in the manuscript, using coloured (red) text. 

Reviewer 1 

1. Spelling out concepts 

First, I think the authors need to do an edit in which they are more careful in how the use the words such as trust, trustworthiness, and 'dimension.' In this regard, the manuscript often suggests that expertise and 'fairness' are dimensions of trust but that's not quite right.

Per the widely accepted 'integrative model of organizational trust' (as well as Fiske's BIAS model, to some extent), it's quite important to distinguish between trustworthiness perceptions and behavioral trust. In this regard, behavioral trust involves actually making oneself vulnerable to a trustee. The behavior part is important. Trustworthiness, in contrast, is a word often used to refer to perceptions and thus ability (or expertise, if you use Fiske's terminology, or competence if you use Hovland's), benevolence (an aspect of warmth for Fiske; part of trustworthiness for Hovland), and integrity (an aspect of warmth for Fiske; part of trustworthiness for Hovland) can be called dimensions of trustworthiness but not dimensions of trust. This distinction is obnoxious but it matters because we really need to make sure we're distinguishing between perceptions/beliefs and behaviors when we do trust-related research.

This distinction is also important because it suggests that just asking people 'how much do you trust X' -- as you do -- will give you really messy data as we have little idea what people are actually thinking about when they respond. Even better is if we ask people more specific questions such as 'to what degree do you think X has expertise in ...' or 'to what degree would you be willing to take the advice of X on Y". Beyond just usage, this edit should thus influence how you understand your results. In any case, I just think you need to clean this up. 

Thanks for this helpful feedback and suggestion. This oversight has been corrected in the manuscript. As suggested by the reviewer, additional text has been added to p. 13-14 (line 261-293) of the paper to succinctly explain and discuss the dimensions of trustworthiness and their relevance to measuring trustworthiness of scientists. Even though our research reveals how different features of scientists change the perceived trustworthiness in a broader context, we certainly agree with the reviewer that it would have been interesting to measure trustworthiness of scientists across different domains of expertise through specific questions (e.g. to what degree do you think Scientist A/B has expertise in X,Y,Z). We will definitely reflect on this in future studies. We now also acknowledged this in the section of discussion p. 24 (line 497-501) 

2. The validity of the method and social desirability bias

“My other ask is that you specifically address the validity/utility of the approach you used. I understand using this type of experiment when you're trying to decide on whether a product with X1 characteristic is more/less appealing than product with X2 characteristics. Indeed, I first learned of the approach in the context of cigarette labeling and it seems reasonable to ask people to pick between two similar options. What I don't get here is why it make sense to ask someone to pick between two scientists. When would someone ever do that unless they were on a hiring committee? And wouldn't doing that make the differences especially salient? Practically, for most times that we ask people to trust scientists, I don't see why we wouldn't be better doing things like presenting systematically randomly different scientists and asking people to rate them on relevant characteristics (e.g., see the work of Shupei Yuan). Practically, I also can't help believing that your results are contaminated by substantial desirability bias.”

Thanks for these comments and concerns. Scholars have indeed argued that conjoint experimental designs can be utilised as an effective measurement tool in measuring socially sensitive attitudes, (e.g. biases against female political candidates (Teele, Kalla and Rosenbluth, 2018), supporting or opposing a specific housing project in one’s neighbourhood (Hankinson, 2018)). In fact, more recently, scholars have empirically tested the claim whether conjoint experimental design has an ability to mitigate social desirability bias (SDB). In this regard, the results confirm that the fully randomised conjoint design like ours can reduce SDB of the sensitive attributes, compared to a standard survey experiment, in which more than one attribute cannot simultaneously be varied randomly across participants (Horiuchi, Markovich, and Yamamoto, 2022). That is, when research subjects are asked to evaluate several attributes simultaneously, they are less concerned that investigators can connect their specific choice to one specific attribute among others. Hence, as we discussed in the section of materials and methods p. 10, conjoint experiments have been used to study various sensitive topics, such as immigration preferences; perceived illegality of ethnic minorities; and attitudes towards asylum seekers, in which respondents, for instance, make hypothetical decisions on asylum admissions and immigration policies. However, as the reviewer pointed out, respondents would not make such decisions in real life scenarios. Hence, even though researchers use such artificial conjoint settings as a proxy to reveal various public preferences, this artificiality may reduce the ecological validity of the research. Given these criticisms, additional text has been added to the manuscript (p. 24-25) to clarify the limitations of our experimental design in relation to its external validity. Lastly, thanks for pointing out the work of Shupei Yuan. Certainly, one can also fully randomise the features of scientists in a conjoint setting to ask people to rate them on relevant characteristics rather than asking for preferences or choices in hypothetical settings. We agree that this may in fact reduce the artificiality in the experimental task. 

- Horiuchi, Y., Markovich, Z., & Yamamoto, T. (2022). Does conjoint analysis mitigate social desirability bias?. Political Analysis, 30(4), 535-549.

- Teele, Dawn Langan, Joshua Kalla and Frances Rosenbluth. 2018. “The Ties That Double Bind: Social Roles and Women’s Underrepresentation in Politics.” American Political Science Review 112(3):525–541.

- Hankinson, Michael. 2018. “When do renters behave like homeowners? High rent, price anxiety, and NIMBYism.” American Political Science Review 112(3):473–493.

3. Limited control over factors in choosing a scientist: A communication perspective

“I'm a little sad that you only varied things over which people have little control. From a communication perspective, I can't do much about my race, gender, or employer. I suppose you could choose a spokesperson for an issue in some cases based on these factors but, in general, the scientist you have is the scientist you get. It is true, however, that a scientist can choose to disclose their political ideology (or religiosity in the case of someone like Katherine Hayhoe or James Watson; or values, more generally, as in the case of of some of Kahan's research), but this isn't exactly a common approach. Training programs (and Hayhoe) often encourage scientists to identify values they share with their audience but I've never heard anyone advocate sharing party ID.”

It is true that our focal point was not derived from a communication perspective when we identified the factors that would help answer our research questions. We have primarily been motivated by the earlier empirical findings that the American public is both culturally and politically divided in their support for science, and the understanding of how the perception of scientists is shaped through their sociodemographic, partisan, and professional characteristics has been somewhat limited. Hence, we created a specific hypothetical scenario where scientists with different attributes ––stereotypes attached to those attributes serve as a cognitive shortcut––can be represented in a local authority for scientific advices instead of being a spokesperson. 

We agree with the reviewer that scientist do not often reveal their political orientation, but people still operate the stereotype of political leanings of scientists as a cognitive cue. For instance, the belief that scientists are mostly liberal has led to conservative criticism of scientific findings and the emergence of conservative organisations that aim to dispute "politically left-leaning" academic science, primarily for the general public (Mann and Schleifer, 2020). As also discussed in the manuscript (p. 9), when scientists address politically polarised scientific issues, people are more likely to perceive the scientists as being a member of the political group associated with that specific issue (Vraga et al. 2018) 

- Vraga E, Myers T, Kotcher J, Beall L, Maibach E. (2018). Scientific risk communication about controversial issues influences public perceptions of scientists’ political orientations and credibility. Royal Society Open Science. 5(2):170505.

- Mann, M., & Schleifer, C. (2020). Love the science, hate the scientists: Conservative identity protects belief in science and undermines trust in scientists. Social Forces, 99(1), 305-332.

4. The differentiation between motivated reasoning and heuristic reasoning

“I think you need to also be a little more careful about how you talk about motivated reasoning. The differentiation between 'motivated' reason and heuristic reasoning seems important here. I'm not sure you're invoking motivated reasoning here inasmuch as you don't have a stimulus that seeks to make a specific identity salient and then ask people to evaluate something using effortful processing. Instead, you're just asking people to make judgements based on heuristic cues, it seems.”

Thank you for this comment. Heuristic reasoning is indeed very closely intertwined with motivated reasoning, and it is therefore important to distinguish between them. In this research, the rationale for choosing motivated reasoning lies within the very framing of the conjoint experimental design. That is, motivated reasoning refers to the use of reasoning and effortful processing to support a particular belief (e.g. whether the scientist with particular qualities is preferrable/trustworthy) rather than using unconscious mental process. We argue that providing respondents with a set of different attributes alongside political affiliations of the scientists’ serves exactly as that kind of stimulus that triggers motivated cognition. Given the variety of the attributes in the conjoint table, evidence (e.g. attributes) may be selectively chosen to fit a pre-determined conclusion on a particular scientist, and it ultimately affects the evaluation of scientists’ credibility and trustworthiness. 

Let’s say, Democrat respondents believe that environmental scientists are in general trustworthy, given their certain qualities, but these respondents can consciously distort their belief if they realise that their political orientations are not congruent with a particular scientist’s political orientation. Moreover, exploring the effects of political partisanship on people’s judgments by invoking motivated cognition has a well-established tradition in social sciences (e.g. see Lewandowsky and Oberauer 2016; Kahan et al. 2017). Following Baron’s definition, on the other hand, heuristics are “methods that are sometimes useful in solving a problem - useful enough to try even when it is not clear that they will help" (Baron 1994 p.70, cited by Lockton 2012 p. 2) and they are more often used in different kind of applications.

- Baron, J. (1994). Thinking and Deciding (2nd edition). Cambridge University Press, Cambridge, UK

- Kahan, D. M., Peters, E., Dawson, E. C., & Slovic, P. (2017). Motivated numeracy and enlightened self-government. Behavioural public policy, 1(1), 54-86.

- Lewandowsky, S., & Oberauer, K. (2016). Motivated rejection of science. Current Directions in Psychological Science, 25(4), 217-222.

- Lockton, D. (2012). Cognitive biases, heuristics and decision-making in design for behaviour change. Heuristics and Decision-Making in Design for Behaviour Change (August 5, 2012).

5. Making sure that figures are self-explanatory 

“The figures need extensive notes that include things like range and the meaning of everything. In this regard, I note that figures generally be understandable without substantial reference to the originating text (at least according to APA).”

As suggested by the reviewer, extensive notes have been added to the figures (p. 18-24) to clarify the definitions of the estimands and descriptive references. 

Reviewer 2 

1. Why are other social science disciplines ignored in the attribute of the field of study? 

“I appreciated the authors’ thorough literature review and argument that informed the selection of the five characteristics of scientists. With regards to the fields of study, I’m curious why the authors did not assess other social/behavioral sciences beyond economics (e.g., psychology, sociology, communication)? These other disciplines would seem germane”

Thanks for questioning this. We certainly agree with the reviewer that it would be insightful to identify how affiliations with other social science disciplines, such as psychology or sociology, impact scientists’ perceived credibility, especially relative to economics. However, in designing our conjoint experiment, we were constrained by the methodological consideration that higher number of levels per attribute may lead to severely under-powered designs (Stefanelli and Lukac 2020). Given our design features, our study satisfactorily reaches the conventional power threshold (≥ 0.80) –the calculations were made through the R Shiny application by Stefanelli and Lukac (2020). This methodological consideration alongside the information on power analysis has now been clarified in the manuscript (p. 13). Being methodologically restricted and willing to test a wide variety of scientific fields urged us to choose only one social science discipline –economics– to represent the whole diverse research area of social/behavioural sciences despite the indisputable heterogeneity in social sciences. Given the research context, we assume that many respondents would have some sort of exposure to economics and its discourses through news outlets, popular magazines, social media. Relatedly, economists frequently offer their knowledge and expertise on a wide range of public policy issues and have seen a steady increase in their influence in scientific enterprises and government, frequently holding high-level positions in institutions (Fourcade et al. 2015). The recent research (Schroder 2022) has also confirmed that economics is considered to be an extremely politicised scientific occupation; this has negative repercussions onto the level of public trust in economists and therefore makes it sound to compare economics with less partisan scientific fields, such as physics.

- Fourcade, M., Ollion, E., & Algan, Y. (2015). The superiority of economists. Revista de Economía Institucional, 17(33), 13-43.

- Stefanelli, A., & Lukac, M. (2020). Subjects, trials, and levels: Statistical power in conjoint experiments.

- Schrøder, T. B. (2022). Don't Tell Me What I Don't Want to Hear! Politicization and Ideological Conflict Explain Why Citizens Have Lower Trust in Climate Scientists and Economists Than in Other Natural Scientists. 

2. Specifying the directionality and strength in the hypothesis 5.2

“For H5.2, can the authors specify the nature of the interaction effect they expected (i.e., “differ” in what way, as far as directionality and strength of association are concerned)?

Many thanks for this feedback. We have specifically avoided to state any directionality and strength in relation to this hypothesis because of its exploratory nature. More precisely, since we were not able to identify established empirical evidence on this potentially unique within-design interaction effect in the literature, we wanted to keep this hypothesis as exploratory rather than confirmatory, in line with the suggestion of open-science practices (Nosek et al. 2018). Indeed, exploratory hypothesis tests tend to be more uncertain and tentative than those of confirmatory hypothesis tests (Rubin and Donkin 2022). Additional texts has been added to the manuscript (p. 10) to incorporate this feedback and clarify our exploratory approach to this hypothesis.

- Nosek, B. A., Ebersole, C. R., DeHaven, A. C., & Mellor, D. T. (2018). The preregistration revolution. Proceedings of the National Academy of Sciences, 115(11), 2600–2606. 

- Rubin, M., & Donkin, C. (2022). Exploratory hypothesis tests can be more compelling than confirmatory hypothesis tests. Philosophical Psychology, 1-29.

3. Proxy alignment rather than direct alignment 

“One could imagine a situation in which the scientist does not affiliate themselves directly with a political party or stance, but instead with being supportive of or opposed to a given behavioral strategy, policy, etc. I realize the authors would not, with these data, be in a position to assess what effects this positioning might have on public trust, but perhaps this is worth considering in future research (and thus could be considered in the discussion; see comment #7). My instinct is that this sort of proxy alignment may be more prevalent than direct alignment/declarations of affiliation with a particular political party or stance (i.e., scientists may have more of an implied vs. overt ideological bent).”

Thank you for this insightful remark. As suggested by the reviewer, the following passage has been added to the manuscript (p .26) to incorporate this feedback into the discussion.

“The second limitation lies in the operationalization of scientists’ political orientation. While measuring it by referring to the well-established categories of political party affiliation (Democrat, Republican, Independent) is appropriate for the US political landscape and conducive to the conjoint design, this is not the only way to do so. Instead of explicitly presenting political affiliation, one could tap into political identification through the proxy alignment of cultural worldview (e.g. see Kahan et al. 2012), or by presenting behavioural cues (e.g. voting in favour of anti-abortion bills) that have a strong linkage to the ideological stance. Further research should explore whether explicit or implicit declaration of scientists’ political affiliation affects their perceived credibility and trustworthiness.” 

4. Clarifying that quota sampling is used in this research

“In the abstract and methods, the authors describe the sample as “representative of U.S. adults,” which is misleading, as Prolific does not maintain a panel created via RDD and/or address-based sampling methods (compared to, say, NORC’s AmeriSpeak panel). It is certainly fair to note the quota sampling.”

We agree with the reviewer. Although we clarify that we use Prolific’s representative quota sampling in the Appendix, describing it as representative in the manuscript without mentioning that it is based on quota sampling can be misleading. This oversight has been corrected in the manuscript. Both the abstract section (p. 1) and method section (p.12) have been revised to clarify that it is a quota sample that is representative of the U.S. population in terms of age, sex, and ethnicity. 

5. The implications of the timing of data collection

“This study was fielded during the early days of the COVID-19 pandemic. This was a rather exceptional time, of course, in which to be evaluating trust in scientists—especially in the U.S., where there is considerable evidence that the pandemic was politicized early and often (see, for example, https://press.princeton.edu/books/hardcover/9780691218991/pandemic-politics and https://read.dukeupress.edu/jhppl/article-abstract/45/6/967/165291/The-Emergence-of-COVID-19-in-the-US-A-Public). Can the authors speak to the implications of the timing of data collection for interpreting and contextualizing study results?”

There has indeed been politicisation of trust in scientists in general in the USA over a fairly long period, when Republicans’ trust has been declining. We reference this material in the article, citing inter alia Gauchat 2012. At a more granular level, we actually know from Pew Research surveys before, during and after the pandemic, that the Republican fall-off in trust in science came after April 2020. (Kennedy et al. (2022) Americans’ Trust in Scientists, Other Groups Declines. In: Pew Research Center Science & Society. Available at: https://www.pewresearch.org/science/2022/02/15/americans-trust-in-scientists-other-groups-declines). The politicisation really got going a bit later on. 

For these reasons, we do not have strong reason to believe that the reactions of US respondents in early March 2020 would have been untypical of responses in less exceptional times. We also include measures of partisanship in our study design, which should capture the main currents of difference between Republican and Democrat respondents. We do, of course, find differences in respondent preferences based on partisan lines in our results. We might hypothesise that these differences would have been even greater had the study been carried out 12 months later. But this is speculative. 

We have added a couple of sentences to make this clear in the methods section (p. 12):

“All questions and question blocks were randomly ordered to avoid spill-over effects. Although the time period of data collection corresponded with the early stages of the Covid-19 pandemic, we do not have reason to believe that in March 2020, attitudes towards scientists had changed much from their pre-pandemic positions. According to surveys from Pew Research, declines in trust in scientists, and in medical scientists in particular, only began after April 2020. (Kennedy et al, 2022). “

6. Is Board of Scientific Councillors used in the U.S. based contexts?

“With regards to DV measurement, I’m not sure that U.S. participants would know what a “Board of Scientific Councillors” is in “their district.” This seems like UK terminology that might not translate. Was this defined for participants? What assurances do we have that participants did in fact understand this reference? Has this measure been used in other U.S.-based contexts?”

We did not give an extended definition of what a board of scientific councilors (BSC) is to respondents, so it is certainly possible that the interpretations could be somewhat heterogeneous. BSCs are definitely an American phenomenon. As we understand it, BSCs can work at both the state and local level, depending on the organization they serve. BSCs may be established by federal or state government agencies to provide advice and guidance on scientific matters specific to that state, such as environmental protection or public health. For example, the CDC (https://www.cdc.gov/cpr/bsc/index.htm) and EPA (https://www.epa.gov/bosc) have these councils. They may also be established by local government agencies, such as a city or county health department, or private organisations to provide advice and guidance on scientific matters specific to that community or mission.

I think it is fair to say that most respondents will not be intimately aware of the role of BSCs but we think that the question contains enough information to provide the essence of what the scientist’s role is. We purposefully avoided giving concrete examples of what a BSC does in order not to associate the concept with any particular form of government that may have influenced perceptions. For example, if we had mentioned that the EPA and NIH have BSCs, or states, this may already have set off reactions based on these organisations or institutions in particular. Hence we simply mentioned ‘local district’ so as not prejudice perception of the scientists themselves in the scenarios.

7. Limitations of the research and implications for journalistic work

“For all the strengths of this study, it is not without limitations, and those should be acknowledged in what is, at present, quite a short discussion section (e.g., see above note about the sample, which has implications for generalizability). Similarly, at present there is a solitary sentence that speaks to the implications of this research for science communication (lines 473-475). Further elaboration here would be helpful: What are the implications for journalistic practice, specifically; for scientists, in their public-facing work; and so forth?”

Thanks for this feedback. As suggested by the reviewer, we have revised the discussion session to acknowledge certain limitations of this research including the implications for generalisability (p. 26-27). In addition, we have added a couple of sentences to elaborate on the implications of our main findings for scientific communication (p. 27-28).

“Specifically, this finding may suggest a need for social scientists to take additional measures to facilitate comprehension of their research by a lay audience. This may include a heightened emphasis on the use of clear and accessible language, the provision of contextual information, and the proactivity in addressing prevalent misconceptions and biases regarding the field. Furthermore, it highlights the need for social scientists to actively engage with the public in order to promote a deeper understanding of their research and to dispel any misconceptions that may exist. In a similar vein, journalists, in their practice of reporting, should be cognizant of the public's perception of the credibility of different fields of science, and critically identify and mitigate any potential biases associated with social sciences in their reporting, through the inclusion of a diverse array of perspectives and scientists in their coverage.” 

To sum up, we are grateful to both reviewers and editorial team for their feedback and suggestions for improving this manuscript. We hope you agree that the revised manuscript is now much improved and suitable for publication.

Best regards,

The authors

---

## [Decision Letter · Decision Letter 1]

4 May 2023

PONE-D-22-23245R1Public perception of scientists: Partisan and non-partisan thinkingPLOS ONE

Dear Dr. Sonmez,

Thank you for submitting your manuscript to PLOS ONE. After careful consideration, we feel that it has merit but does not fully meet PLOS ONE’s publication criteria as it currently stands. Therefore, we invite you to submit a revised version of the manuscript that addresses the points raised during the review process.

We look forward to receiving your revised manuscript.

Kind regards,

Celia Andreu-Sánchez

Academic Editor

PLOS ONE

Journal Requirements:

Reviewers' comments:

Reviewer's Responses to Questions

**Comments to the Author**

1. If the authors have adequately addressed your comments raised in a previous round of review and you feel that this manuscript is now acceptable for publication, you may indicate that here to bypass the “Comments to the Author” section, enter your conflict of interest statement in the “Confidential to Editor” section, and submit your "Accept" recommendation.

Reviewer #3: All comments have been addressed

Reviewer #4: (No Response)

2. Is the manuscript technically sound, and do the data support the conclusions?

Reviewer #3: Yes

Reviewer #4: Yes

3. Has the statistical analysis been performed appropriately and rigorously? 

Reviewer #3: Yes

Reviewer #4: Yes

4. Have the authors made all data underlying the findings in their manuscript fully available?

Reviewer #3: Yes

Reviewer #4: Yes

5. Is the manuscript presented in an intelligible fashion and written in standard English?

Reviewer #3: Yes

Reviewer #4: Yes

6. Review Comments to the Author

Reviewer #3: You did an adequate job responding to the suggestions of the reviewers. However, the quality of the figures should be addressed before publication.

Reviewer #4: Thank you for the opportunity to review “Public perceptions of scientists: partisan and non-partisan thinking.” I was not an initial reviewer on this piece, but I appreciated the previous reviewers’ very thoughtful suggestions and the authors’ comprehensive work to be responsive. I think it’s an important analysis and a solid study design that contributes new insights about how the public understands scientists and the heterogeneity in perspectives attributable to different characteristics of both scientists and of the public evaluating scientists. I have mainly minor comments, and some reinforcing of previous reviewers’ comments.

First, I’m not sure the title is accurate – or at least, does not convey the main points of this paper. I wasn’t sure what “partisan and non-partisan thinking” meant when I started the paper, and I’m still not sure this well-captures the paper. It reduces the paper to the partisan cues/heuristics and I think doesn’t convey much of what the paper does. Perhaps “Public perception of scientists: partisan and non-partisan characteristics contributing to perceptions” or something similar?

In some of the rationale and background supporting why the authors tested different characteristics of scientists, they conflate attributes of the scientists with those same attributes (or stereotypes related to those attributes) among the general public. This is most salient in the paragraph about race (111-122) where some of the justifications are about racialized stereotypes about scientists and some, such as racial differences in science literacy, are about the general public. I suggest authors edit this paragraph accordingly to be more clear that they are referring to racialized perceptions of scientists.

Also in the background (lines 168-176), is an alternative explanation for why some scientific fields might be regarded with less trust or credibility be how applied the science is into policy and politics? I think vaccination, infectious disease, climate change can also be characterized by how closely adjacent those fields are to relevant policy decisions that get embroiled in politics (mandates, restrictions, and taxes or other consumer restrictions) and not only how “permeated” they are by conspiratorial thinking.

I know the interaction between partisanship and institution/place of work is only exploratory, but I still think it needs more explanation. For instance, the authors note that “Republicans and Democrats have contrasting views on institutions” (line 201) but do not explain what these views are.

The study was field in March of 2020. I know one of the previous reviewers asked the author to spell out how this timing might have been related to trust and the authors note that trust declines began later. However, there were other elements of partisan gaps in views about the pandemic that emerged as early as February and early March (see https://fivethirtyeight.com/features/how-concerned-are-americans-about-coronavirus-so-far/). I would like to see the authors engage a bit more about how the salience of the emerging pandemic might shape perhaps how respondents viewed reference to medical science in particular as a field, or to whether and how the stage of the pandemic could have affected survey response or other aspects of survey administration. (Please also note the actual dates that the study was fielded, not just “March of 2020”).

Like Reviewer 2, I too was surprised by the language of “Board of Scientific Councillors in your district.” This is not a concept I have ever heard of (as a U.S. citizen who studies science communication!) so I’m doubtful that most respondents had heard of this or knew what they were responding about. Further, the concept of “district” is not commonly used in the U.S. to refer to local government (as opposed to county, municipality, or town/city). District normally refers to schools, not other governmental entities. While I see the authors’ response to the initial reviewer who raised this, I would push back gently and suggest that there may be a limitation in the study design wherein there could be some unmeasured error in how respondents understood this concept that they were rating. Would suggest more acknowledgement or a footnote about this concern.

For the 3 key dependent variables, please state the measurement of the outcomes outright. Authors say they are measured on a Likert scale “where 7 is strongly trust and 1 is strongly mistrust” but this measurement does not make sense for the first DV (“where would you place your assessment”) nor the second “How much would you agree..”). It would be clearest to just note the response options for each item.

In the limitations, I would like to see the authors emphasize Reviewer 1’s very important concern about ecological validity more forcefully (lines 527-528). They acknowledge rather obliquely that the task is artificial, but I think they should acknowledge more explicitly that the task they are asking respondents to do (select a scientist to serve in a particular role) is something the general public will rarely if ever actually do. I think the tradeoffs between the validity of this task and the causal inference of the design are justified, but it does bear more explicit discussion.

Overall I learned a lot from this study and found the findings interesting, thought-provoking, and well supported by the rigorous study design. I also appreciated the comprehensive literature review which I suspect will also be useful for future research teams when the study is published.

Minor points:

I found some examples of awkward language and language that could be more concise. Of course, the authors can decide not to edit these, but just flagging:

Line 73 – “public preference for science is unequally distributed” – unequal distribution sounds like a moral claim, when I think the authors just mean “heterogeneous based on…”

Line 77 – the reference to messages that scientists communicate is confusing, as this study does not address messaging at all. And yet in lines 77 through the Fauci example, the authors describe messages that trigger reactions based on audience predispositions. I would cut or edit this content as it made me think the paper was going to address communication and source x message interactions.

Line 95-96 – “a considerable fraction of stereotypes concerning sex roles conveys a denigrating message about women and often denies that they possess certain traits” – this is awkward and the subject-verb agreement is off (i.e., stereotypes convey messages; stereotypes deny that they possess); I would rephrase.

Line 133 – “depending on whether they are state, industry, or academic” – should “state” be government, or is the reference to the U.S. states?

Line 146 – “are open to being mistrusted from a different perspective.” This took me a while to understand. Perhaps another way to word this is that “scientists working for government…potentially face mistrust from a different perspective”?

Lines 165-167 – can authors clarify what they mean by autonomy-heteronomy?

Figures – like Reviewer 1, I want to make sure that the figures have comprehensive notes attached. For instance, I know what the Partisan column is referring to in the first figure, and what the non-partisan column is, but what is “Nonpartisan-Partisan”? Double checking that there are complete notes and legends for all exhibits will be important, as this was hard for me to assess in how the figures appeared in the reviewer copy.

7. PLOS authors have the option to publish the peer review history of their article (what does this mean?). If published, this will include your full peer review and any attached files.

Reviewer #3: No

Reviewer #4: No

---

## [Author Response · Author response to Decision Letter 1]

19 May 2023

Dear Editors and Reviewers,

Thank you for the invitation to revise and resubmit our manuscript entitled “Public Perception of Scientists: Partisan and Non-partisan Thinking” (PONE-D-22-23245). We are grateful to both reviewers and editors for their detailed feedback and suggestions for minor revisions. We have edited the manuscript in regards to actionable suggestions and further comments. In line with the journal requirements, we have also reviewed our reference list to ensure that it is complete and correct. 

As requested, we detail how we have addressed each of the comments and acted upon the suggestions. We have highlighted specific changes in the manuscript, using coloured (red) text. 

Reviewer 3:

1. You did an adequate job responding to the suggestions of the reviewers. However, the quality of the figures should be addressed before publication.

Thanks for the feedback. Before resubmitting, we rechecked the figures and also used PLOS ONE’s suggested platform “PACE” to make sure the quality of the figures. 

Reviewer 4:

1. First, I’m not sure the title is accurate – or at least, does not convey the main points of this paper. I wasn’t sure what “partisan and non-partisan thinking” meant when I started the paper, and I’m still not sure this well-captures the paper. It reduces the paper to the partisan cues/heuristics and I think doesn’t convey much of what the paper does. Perhaps “Public perception of scientists: partisan and non-partisan characteristics contributing to perceptions” or something similar?

Thanks for this feedback. Based on the suggestion, we revised the title to better convey the main points of the study. The updated title is “Public perception of scientists: Experimental evidence on the role of sociodemographic, partisan, and professional characteristics” 

2. In some of the rationale and background supporting why the authors tested different characteristics of scientists, they conflate attributes of the scientists with those same attributes (or stereotypes related to those attributes) among the general public. This is most salient in the paragraph about race (111-122) where some of the justifications are about racialized stereotypes about scientists and some, such as racial differences in science literacy, are about the general public. I suggest authors edit this paragraph accordingly to be more clear that they are referring to racialized perceptions of scientists.

This paragraph has been edited to clarify that we are interested in racialised perceptions of scientists.

3. Also in the background (lines 168-176), is an alternative explanation for why some scientific fields might be regarded with less trust or credibility be how applied the science is into policy and politics? I think vaccination, infectious disease, climate change can also be characterized by how closely adjacent those fields are to relevant policy decisions that get embroiled in politics (mandates, restrictions, and taxes or other consumer restrictions) and not only how “permeated” they are by conspiratorial thinking.

Thank you for this insightful comment. We agree with the reviewer. A couple of sentences have been added to the manuscript (p. 8 lines 173-177) to incorporate this feedback into the background. 

4. I know the interaction between partisanship and institution/place of work is only exploratory, but I still think it needs more explanation. For instance, the authors note that “Republicans and Democrats have contrasting views on institutions” (line 201) but do not explain what these views are.

This oversight has been corrected in the manuscript. As suggested by the reviewer, additional text has been added to (p. 10 lines 208-209) to flesh out those views. 

5. The study was field in March of 2020. I know one of the previous reviewers asked the author to spell out how this timing might have been related to trust and the authors note that trust declines began later. However, there were other elements of partisan gaps in views about the pandemic that emerged as early as February and early March (seehttps://fivethirtyeight.com/features/how-concerned-are-americans-about-coronavirus-so-far/). I would like to see the authors engage a bit more about how the salience of the emerging pandemic might shape perhaps how respondents viewed reference to medical science in particular as a field, or to whether and how the stage of the pandemic could have affected survey response or other aspects of survey administration. (Please also note the actual dates that the study was fielded, not just “March of 2020”).

Thanks for this feedback. Based on the reviewer’s suggestion, we first clarified the actual data collection dates in the manuscript (p. 12 lines 249-250) and then how the partisan gap in the confidence in the U.S. health care system in dealing with the coronavirus may have an impact on respondents’ confidence in medical science as a field (p. 13 lines 260-268). More precisely, we highlighted that there was a partisan gap in the confidence in the U.S. health care system in dealing with the coronavirus. That is, in early March 2020, a vast majority (87%) of Republicans had confidence in the U.S. health care system to handle the response to the coronavirus, while a slight majority (53%) of Democrats had this positive view (Malloy and Schwartz 2020) (This new reference has been added to the list). This partisan heterogeneity in turn may have a differential impact on respondents’ confidence in medical science and potentially scientists in this scientific field. Hence, it is crucial to include measures of respondent’s partisanship in our study design, which can capture the main currents of difference between Republican and Democrat respondents. 

- Tim Malloy, Doug Schwartz. Biden crushes Sanders in democratic race, Quinnipiac University national poll finds; More disapprove of Trump’s response to coronavirus [Internet]. Quinnipiac University Poll; 2020. Available from: https://poll. qu. edu/national/release-detail.

6. Like Reviewer 2, I too was surprised by the language of “Board of Scientific Councillors in your district.” This is not a concept I have ever heard of (as a U.S. citizen who studies science communication!) so I’m doubtful that most respondents had heard of this or knew what they were responding about. Further, the concept of “district” is not commonly used in the U.S. to refer to local government (as opposed to county, municipality, or town/city). District normally refers to schools, not other governmental entities. While I see the authors’ response to the initial reviewer who raised this, I would push back gently and suggest that there may be a limitation in the study design wherein there could be some unmeasured error in how respondents understood this concept that they were rating. Would suggest more acknowledgement or a footnote about this concern.

Thanks for raising this concern. We have added a couple of sentences (p. 28 lines 569-573) to clarify that the specific language used to define the scientific board in the task (i.e. Board of Scientific Councillors in their district) may have been misunderstood by the U.S. participants, because as the reviewer suggests, the concept is not widely known. We acknowledge that this limitation may lead to some unmeasured error in operationalising our dependent variables.

7. For the 3 key dependent variables, please state the measurement of the outcomes outright. Authors say they are measured on a Likert scale “where 7 is strongly trust and 1 is strongly mistrust” but this measurement does not make sense for the first DV (“where would you place your assessment”) nor the second “How much would you agree..”). It would be clearest to just note the response options for each item.

This oversight has been corrected in the manuscript (p. 16). 

8. In the limitations, I would like to see the authors emphasize Reviewer 1’s very important concern about ecological validity more forcefully (lines 527-528). They acknowledge rather obliquely that the task is artificial, but I think they should acknowledge more explicitly that the task they are asking respondents to do (select a scientist to serve in a particular role) is something the general public will rarely if ever actually do. I think the trade-offs between the validity of this task and the causal inference of the design are justified, but it does bear more explicit discussion.

The discussion part (p.27-28 lines 565-569) has been revised to explicitly acknowledge that in our study, respondents take a decision task in which they are asked to elect a scientist, yet the general public is less likely to get involved in this type of elections in real-world context. The artificial setting therefore undermines the ecological validity of the findings.

Minor points: 

Line 73 – “public preference for science is unequally distributed” – unequal distribution sounds like a moral claim, when I think the authors just mean “heterogeneous based on…”

Thanks. We agree with the reviewer. We changed it accordingly. 

Line 77 – the reference to messages that scientists communicate is confusing, as this study does not address messaging at all. And yet in lines 77 through the Fauci example, the authors describe messages that trigger reactions based on audience predispositions. I would cut or edit this content as it made me think the paper was going to address communication and source x message interactions.

Based on the reviewer’s suggestion, we cut the content through Fauici’s example to avoid confusions (p. 4 line 79). 

Line 95-96 – “a considerable fraction of stereotypes concerning sex roles conveys a denigrating message about women and often denies that they possess certain traits” – this is awkward and the subject-verb agreement is off (i.e., stereotypes convey messages; stereotypes deny that they possess); I would rephrase.

Thanks. We have rephrased this sentence to clarify the point (p. 5 lines-94-95). 

Line 133 – “depending on whether they are state, industry, or academic” – should “state” be government, or is the reference to the U.S. states?

Yes, it should have been “government”. We have corrected this (p. 6 line 134). 

Line 146 – “are open to being mistrusted from a different perspective.” This took me a while to understand. Perhaps another way to word this is that “scientists working for government…potentially face mistrust from a different perspective”?

We agree with the reviewer. We have revised the sentence (p. 7 lines 147-148). 

Lines 165-167 – can authors clarify what they mean by autonomy-heteronomy?

Sure. It’s a Bourdieusian concept, which means that people differentiate between how knowledge relates to broader society, whether the audience is exclusively other knowledge producers (i.e. autonomous), or if economic and political authorities outside the knowledge domain are also the audience (i.e. heteronomous). However, we have decided to exclude this concept, since it creates ambiguity for the journal’s broad readership. 

Figures – like Reviewer 1, I want to make sure that the figures have comprehensive notes attached. For instance, I know what the Partisan column is referring to in the first figure, and what the non-partisan column is, but what is “Nonpartisan-Partisan”? Double checking that there are complete notes and legends for all exhibits will be important, as this was hard for me to assess in how the figures appeared in the reviewer copy.

Thanks. We double-checked that notes are complete and clarified that the first panel (Nonpartisan) shows the results of the conjoint profiles without the partisan profiles. The second panel (Partisan) shows the results of the conjoint profiles with the partisan profiles. The third panel (Nonpartisan-Partisan) shows the differences between the partisan and nonpartisan profiles.

Overall, we are grateful to all reviewers and editorial team for their feedback and suggestions for improving this manuscript. We hope you agree that the revised manuscript is now much improved and suitable for publication.

Best regards, 

The authors

---

## [Decision Letter · Decision Letter 2]

8 Jun 2023

Public perception of scientists: Experimental evidence on the role of sociodemographic, partisan, and professional characteristics

PONE-D-22-23245R2

Dear Dr. Sonmez,

We’re pleased to inform you that your manuscript has been judged scientifically suitable for publication and will be formally accepted for publication once it meets all outstanding technical requirements.

Kind regards,

Celia Andreu-Sánchez

Academic Editor

PLOS ONE

Additional Editor Comments (optional):

Reviewers' comments:

Reviewer's Responses to Questions

**Comments to the Author**

1. If the authors have adequately addressed your comments raised in a previous round of review and you feel that this manuscript is now acceptable for publication, you may indicate that here to bypass the “Comments to the Author” section, enter your conflict of interest statement in the “Confidential to Editor” section, and submit your "Accept" recommendation.

Reviewer #3: All comments have been addressed

Reviewer #4: All comments have been addressed

2. Is the manuscript technically sound, and do the data support the conclusions?

Reviewer #3: Yes

Reviewer #4: Yes

3. Has the statistical analysis been performed appropriately and rigorously? 

Reviewer #3: Yes

Reviewer #4: Yes

4. Have the authors made all data underlying the findings in their manuscript fully available?

Reviewer #3: Yes

Reviewer #4: Yes

5. Is the manuscript presented in an intelligible fashion and written in standard English?

Reviewer #3: Yes

Reviewer #4: Yes

6. Review Comments to the Author

Reviewer #3: I think you have addressed the reviewers' comments adequately. For future research on this topic, it might be worth incorporating Douglas and Wildavsky's Cultural Theory, which has been used mostly in the risk perception literature, but I think would also help predict the public's perceptions of scientists.

Reviewer #4: The authors have done an excellent job responding to my concerns and suggestions and the paper is improved as a result.

7. PLOS authors have the option to publish the peer review history of their article (what does this mean?). If published, this will include your full peer review and any attached files.

Reviewer #3: No

Reviewer #4: No
